# Salsa Fresca: Angular Embeddings and Pre-Training for ML Attacks on Learning With Errors

**Samuel Stevens**  *stevens.994@osu.edu*
*The Ohio State University*

**Emily Wenger**  *emily.wenger@duke.edu*
*Duke University*

**Cathy Yuanchen Li**  *yuanchen@uchicago.edu*
*University of Chicago*

**Niklas Nolte**  *nolte@meta.com*
*Meta AI Research*

**Eshika Saxena**  *eshika@meta.com*
*Meta AI Research*

**François Charton**  *fcharton@meta.com*
*Meta AI Research*

**Kristin E. Lauter**  *klauter@meta.com*
*Meta AI Research*

**Reviewed on OpenReview:** *https://openreview.net/forum?id=w4nd5695sq*

## Abstract

Learning with Errors (LWE) is a hard math problem underlying recently standardized post-quantum cryptography (PQC) systems for key exchange and digital signatures (Chen et al., 2022). Prior work (Wenger et al., 2022; Li et al., 2023a;b) proposed new machine learning (ML)-based attacks on LWE problems with small, sparse secrets, but these attacks require millions of LWE samples to train on and take days to recover secrets. We propose three key methods—better preprocessing, angular embeddings and model pre-training—to improve these attacks, speeding up preprocessing by $25\times$ and improving model sample efficiency by $10\times$. We demonstrate for the first time that pre-training improves and reduces the cost of ML attacks on LWE. Our architecture improvements enable scaling to larger-dimension LWE problems: this work is the first instance of ML attacks recovering sparse binary secrets in dimension $n = 1024$, the smallest dimension used in practice for homomorphic encryption applications of LWE where sparse binary secrets are proposed Lauter et al. (2011), albeit for larger modulus $q$. The ML-based approach is the only attack which has successfully recovered secrets for these parameters.

## 1 Introduction

Lattice-based cryptography was recently standardized by the US National Institute of Standards and Technology (NIST) in the 5-year post-quantum cryptography (PQC) competition (Chen et al., 2022). Lattice-based schemes are believed to be resistant to attacks by both classical and quantum computers. Given their importance for the future of information security, verifying the security of these schemes is critical, especially when special parameter choices are made such as binary, ternary, and/or sparse secrets. Both the NIST standardized schemes and homomorphic encryption (HE) rely on the hardness of the "Learning

Table 1: **Best attack results** for LWE problems in dimensions $n$ (higher is harder), modulus $q$ (lower is harder), and Hamming weights $h$ (higher is harder). We recover secrets for $n = 1024$ for the first time in ML-based LWE attacks and reduce total attack time for $n = 512, \log_2 q = 41$ to 50 hours (with full CPU parallelization).

| n | log$_2$q | highest h | LWE $(\mathbf{A}, \mathbf{b})$ matrices needed | preprocessing time (hrs/CPU/matrix) | training time (hrs) | total time (hrs) |
|---|---|---|---|---|---|---|
| 512 | 41 | 44 | 1955 | 13.1 | 36.9 | 50.0 |
| 768 | 35 | 9 | 1302 | 12.4 | 14.8 | 27.2 |
| 1024 | 50 | 13 | 977 | 26.0 | 47.4 | 73.4 |

with Errors" (Regev, 2005, LWE) problem. NIST schemes standardize small secrets (binomial), and the HE standard includes binary and ternary secrets Albrecht et al. (2021); Bossuat et al. (2024), with sparse versions used in practice Cheon et al. (2017).

The LWE problem is defined as follows: in dimension $n$, the secret $\mathbf{s} \in \mathbb{Z}_q^n$ is a vector of length $n$ with integer entries modulo $q$. Let $\mathbf{A} \in \mathbb{Z}_q^{m \times n}$ be a uniformly random matrix with $m$ rows, and $\mathbf{e} \in \mathbb{Z}_q^m$ an error vector sampled from a narrow Gaussian $\chi_e$ (see 2 for notation summary). The goal is to find $\mathbf{s}$ given $\mathbf{A}$ and $\mathbf{b}$, where $\mathbf{b} = \mathbf{A} \cdot \mathbf{s} + \mathbf{e} \mod q$. The hardness of this problem depends on the parameter choices: $n$, $q$, and the secret and error distributions. Many HE implementations use sparse, small secrets to improve efficiency and functionality, where all but $h$ entries of $\mathbf{s}$ are zero (so $h$ is the Hamming weight of $\mathbf{s}$). The non-zero elements have a limited range of values: 1 for binary secrets, 1 and $-1$ for ternary secrets. Sparse binary or ternary secrets allow for fast computations, since they replace the $n$-dimensional scalar product $\mathbf{A} \cdot \mathbf{s}$ by $h$ sums. However, binary and ternary secrets might be less secure Bai & Galbraith (2014).

Most attacks on LWE rely on lattice reduction techniques, such as `LLL` or `BKZ` Lenstra et al. (1982); Schnorr (1987); Chen & Nguyen (2011), which recover $\mathbf{s}$ by finding short vectors in a lattice constructed from $\mathbf{A}, \mathbf{b}$ and $q$ Ajtai (1996); Chen et al. (2020). `BKZ` attacks scale poorly to large dimension $n$ and small moduli $q$ Albrecht et al. (2015).

ML-based attacks on LWE were first proposed in Wenger et al. (2022); Li et al. (2023a;b), inspired by viewing LWE as a linear regression problem on a discrete torus. These attacks train small transformer models (Vaswani et al., 2017) to extract a secret from eavesdropped LWE samples ($\mathbf{A}$, $\mathbf{b}$), using lattice-reduction for preprocessing. Although Li et al. solves medium-to-hard small sparse LWE instances, for example, dimension $n = 512$, the approach is bottlenecked by preprocessing and larger dimensions used in HE schemes, such as $n = 1024$. In this work, we introduce several improvements to Li et al.'s ML attack, enabling secret recovery for harder LWE problems in less time. Our main contributions are:

- **25× faster pre-processing** using `Flatter` (Ryan & Heninger, 2023) and interleaving with polishing (Charton et al., 2024) and `BKZ2.0` (Chen & Nguyen, 2011) (see §3).

- An **encoder-only transformer architecture, coupled with an angular embedding for model inputs.** This reduces the model's logical and computational complexity and halves the input sequence length, significantly improving model performance (see §4).

- The **first use of pre-training** for LWE to improve sample efficiency for ML attacks, further reducing preprocessing cost by **10×** (see §5).

Overall, these improvements in both preprocessing and modeling allow us to recover secrets for harder instances of LWE, i.e. higher dimension and lower modulus in less time and with fewer computing resources. A summary of our main results can be found in Table 1. Although no prior LWE attack has been shown to succeed experimentally at the parameter settings we consider, in §6 we contextualize our performance by presenting theoretical estimates for other LWE attacks. The ML-based approach is the only attack which has successfully recovered secrets for these parameters.

Table 2: **Notation used in this work.**

| Symbol | Description |
|---|---|
| $(\mathbf{A}, \mathbf{b})$ | LWE matrix/vector pair, with $\mathbf{b} = \mathbf{A} \cdot \mathbf{s} + \mathbf{e}$. |
| $(\mathbf{a}, b)$ | An LWE vector/integer pair, one row of $(\mathbf{A}, \mathbf{b})$. |
| $q$ | Modulus of the LWE problem. |
| $\mathbf{s}$ | The (unknown) true secret. |
| $h$ | Number of nonzero bits in secret. |
| $\mathbf{e}$ | Error vector, drawn from distribution $\chi_e$ |
| $\sigma_e$ | Standard deviation of the $\chi_e$. |
| $n$ | Problem dimension (the dimension of $\mathbf{a}$ and $\mathbf{s}$) |
| $t$ | The total number of LWE samples available |
| $m$ | # LWE samples in each subset during reduction |
| $\mathbf{s}^*$ | Candidate secret, not necessarily correct |
| $\mathbf{R}$ | Matrix computed to reduce the coordinates of $\mathbf{A}$. |
| $\rho$ | Preprocessing reduction factor; the ratio $\frac{\sigma(\mathbf{RA})}{\sigma(\mathbf{A})}$ |

## 2 Context and Attack Overview

Wenger et al. (2022) and Li et al. (2023a;b) demonstrated the feasibility of ML-based attacks on LWE. Li et al. (2023a) has 2 parts: 1) data preprocessing using lattice reduction techniques; 2) model training interleaved with regular calls to a secret recovery routine, using a trained model as a cryptographic distinguisher to guess the secret. In this section we provide an overview of ML-based attacks in prior work.

### 2.1 Attack Part 1: LWE data preprocessing

The attack assumes $t = 4n$ initial LWE samples $(\mathbf{a}, b)$ (rows of $(\mathbf{A}, \mathbf{b})$) with the same secret are available. Sampling $m \leq n$ of the $4n$ initial samples without replacement, matrices $\mathbf{A} \in \mathbb{Z}_q^{m \times n}$ with vectors $\mathbf{b} \in \mathbb{Z}_q^m$ are constructed.

The preprocessing step strives to reduce the norm of the rows of $\mathbf{A}$ by applying a carefully selected integer linear operator $\mathbf{R}$. Because $\mathbf{R}$ is linear with integer entries, the transformed pairs $(\mathbf{RA}, \mathbf{Rb})$ mod $\mathbf{q}$ are also LWE pairs with the same secret, albeit larger error. In practice, $\mathbf{R}$ is found by performing lattice reduction on the $(m + n) \times (m + n)$ matrix $\mathbf{\Lambda} = \begin{bmatrix} 0 & q \cdot \mathbf{I}_n \\ \omega \cdot \mathbf{I}_m & \mathbf{A} \end{bmatrix}$, and finding linear operators $\begin{bmatrix} \mathbf{C} & \mathbf{R} \end{bmatrix}$ such that the norms of $\begin{bmatrix} \mathbf{C} & \mathbf{R} \end{bmatrix} \mathbf{\Lambda} = \begin{bmatrix} \omega \cdot \mathbf{R} & \mathbf{RA} + q \cdot \mathbf{C} \end{bmatrix}$ are small. This achieves a reduction of the norms of the entries of $\mathbf{RA}$ mod $q$, but also increases the error in the calculation of $\mathbf{Rb} = \mathbf{RA} \cdot \mathbf{s} + \mathbf{Re}$, making secret recovery more difficult. Although ML models can learn from noisy data, too much noise will make the distribution of $\mathbf{Rb}$ uniform on $[0, q)$ and inhibit learning. The parameter $\omega$ controls the trade-off between norm reduction and error increase. Reduction strength is measured by $\rho = \frac{\sigma(\mathbf{RA})}{\sigma(\mathbf{A})}$, where $\sigma$ denotes the mean of the standard deviations of the rows of $\mathbf{RA}$ and $\mathbf{A}$.

Li et al. (2023a) use BKZ Schnorr (1987) for lattice reduction  Li et al. (2023b) improves the reduction time by 45× via a modified definition of the $\Lambda$ matrix and by interleaving `BKZ2.0` Chen & Nguyen (2011) and `polish` Charton et al. (2024) (see Appendix F).

This preprocessing step produces many $(\mathbf{RA}, \mathbf{Rb})$ pairs that can be used to train models. Individual rows of $\mathbf{RA}$ and associated elements of $\mathbf{Rb}$, denoted as reduced LWE samples $(\mathbf{Ra}, \mathbf{R}b)$ with some abuse of notation, are used for model training. Both the subsampling of $m$ samples from the original $t$ LWE samples and the reduction step are done repeatedly and in parallel to produce 4 million reduced LWE samples, providing the data needed to train the model.

## 2.2 Attack Part 2: Model training and secret recovery

With 4 million reduced LWE samples $(\mathbf{R}\mathbf{a}, \mathbf{R}b)$, a transformer is trained to predict $\mathbf{R}b$ from $\mathbf{R}\mathbf{a}$. For simplicity, and without loss of generality, we will say the transformer learns to predict $b$ from $\mathbf{a}$. Li et al. train encoder-decoder transformers (Vaswani et al., 2017) with shared layers (Dehghani et al., 2019). Inputs and outputs consist of integers that are split into two tokens per integer by representing them in a large base $B = \frac{q}{k}$ with $k \approx 10$ and binning the lower digit to keep the vocabulary small as $q$ increases. Training is supervised and minimizes a cross-entropy loss.

The key intuition behind ML-attacks on LWE is that to predict $b$ from $\mathbf{a}$, the model must have learned the secret $\mathbf{s}$. We extract the secret from the model by comparing model predictions for two vectors $\mathbf{a}$ and $\mathbf{a}'$ which only differ on one entry. We expect the difference between the model's predictions for $b$ and $b'$ to be small (of the same magnitude as the error) if the corresponding bit of $\mathbf{s}$ is zero, and large if it is non-zero. Repeating the process on all $n$ positions yields a guess for the secret.

For ternary secrets, Li et al. (2023b) introduce a two-bit distinguisher, which leverages the fact that if secret bits $s_i$ and $s_j$ have the same value, adding a constant $K$ to inputs at both these indices should induce similar predictions. Thus, if $\mathbf{u}_i$ is the $i^{th}$ basis vector and $K$ is a random integer, we expect model predictions for $\mathbf{a} + K\mathbf{u}_i$ and $\mathbf{a} + K\mathbf{u}_j$ to be the same if $s_i = s_j$. After using this pairwise method to determine whether the non-zero secret bits have the same value, Li et al. classify them into two groups. With only two ways to assign 1 and $-1$ to the groups of non-zero secret bits, this produces two secret guesses.

Wenger et al. (2022) test a secret guess $\mathbf{s}^*$ by computing the residuals $b - \mathbf{a} \cdot \mathbf{s}^*$ over the $4n$ initial LWE sample. If $\mathbf{s}^*$ is correct, the standard deviation of the residuals will be close to $\approx \sigma_e$. Otherwise, it will be close to the standard deviation of a uniform distribution over $\mathbb{Z}_q$: $q/\sqrt{12}$.

For a given dimension, modulus and secret Hamming weight, the performance of ML attacks vary from one secret to the next. Li et al. (2023b) observe that the difficulty of recovering a given secret $\mathbf{s}$ from a set of reduced LWE samples $(\mathbf{a}, b)$ depends on the distribution of the scalar products $\mathbf{a} \cdot \mathbf{s}$. If a large proportion of these products remain in the interval $(-q/2, q/2)$ (assuming centering) even without a modulo operation, the problem is similar enough to linear regression that the ML attack will usually recover the secret. Li et al. introduce the statistic **NoMod**: the proportion of scalar products in the training set having this property. They demonstrate that large **NoMod** strongly correlates with likely secret recoveries for the ML-attack.

## 2.3 Improving upon prior work

Li et al. (2023b) recover binary and ternary secrets, for $n = 512$ and $\log_2 q = 41$ LWE problems with Hamming weight $\leq 63$, in 36 days, using 4,000 CPUs and 1 GPU ((Li et al., 2023b, Table 1)). Most of the computing resources are needed in the preprocessing stage: reducing one $m \times n$ $\mathbf{A}$ matrix takes 35 days, and 4000 matrices must be reduced to build a training set of 4 million examples. This suggests two directions for improving the attack performance. First, introducing fast alternatives to `BKZ2.0` may shorten the time required to reduce one matrix. Second, minimizing the number of samples needed to train the models would reduce the number of CPUs needed for the preprocessing stage.

Another crucial goal is scaling to larger dimensions $n$. The smallest standardized dimension in the HE Standard Albrecht et al. (2021) is $n = 1024$. At present, ML attacks are limited by their preprocessing time and the length of the input sequences. The attention mechanism used in transformers is quadratic in the sequence length, and Li et al. (2023b) encodes $n$ dimensional inputs with $2n$ tokens. More efficient encoding would cut down on transformer processing speed and memory consumption quadratically.

## 2.4 Parameters and settings in our work

Before presenting our innovations and results, we briefly discuss the LWE settings considered in our work. LWE problems are parameterized by the modulus $q$, the secret dimension $n$, the secret distribution $\chi_{\mathbf{s}}$ (sparse binary/ternary) and the hamming weight $h$ of the secret (the number of non-zero entries). Table 3 specifies the LWE problem settings we attack. Proposals for LWE parameter settings in homomorphic encryption suggest using $n = 1024$ with sparse secrets (as low as $h = 64$), albeit with smaller $q$ than we consider Curtis

Table 3: **LWE parameters attacked in our work.** For all settings, we attack both binary and ternary secret distributions $\chi_{\mathbf{s}}$.

| $n$ | $\log_2 q$ | $h$ |
|------|------------|------------------|
| 512 | 41 | $50 \leq h \leq 70$ |
| 768 | 35 | $5 \leq h \leq 15$ |
| 1024 | 50 | $5 \leq h \leq 15$ |

Table 4: **Reduction performance and median time to reduce one matrix for Li et al. (2023b) vs. our work.** Li et al.'s method fails for $n > 512$ on our compute cluster.

| $n$ | $\log_2 q$ | $\rho$ | CPU · hours · matrix | |
|------|------------|--------|------------------|------|
| | | | Li et al. (2023b) | Ours |
| 512 | 41 | 0.41 | $\approx 350$ | 13.1 |
| 768 | 35 | 0.71 | N/A | 12.4 |
| 1024 | 50 | 0.70 | N/A | 26.0 |

Table 5: **Tradeoff between reduction quality and error is controlled by $\omega$.** $n = 1024$, $\log_2 q = 50$.

| $\omega$ | 1 | 3 | 5 | 7 | 10 | 13 |
|------|-------|-------|-------|-------|-------|-------|
| $\rho$ | 0.685 | 0.688 | 0.688 | 0.694 | 0.698 | 0.706 |
| $\|\mathbf{R}\|/q$ | 0.341 | 0.170 | 0.118 | 0.106 | 0.075 | 0.068 |

& Player (2019); Albrecht (2017). Thus, it is important to show that ML attacks can work in practice for dimension $n = 1024$ if we hope to attack sparse secrets in real-world settings. Similar to developing attacks on reduced-round AES, we develop attacks on LWE problems with larger $q$ and lower $h$ than in practice. Future work should further improve $q$ and $h$ towards concrete homomorphic encryption proposals.

The LWE error distribution remains the same throughout our work: rounded Gaussian with $\sigma = 3$ (following Albrecht et al. (2021)). Table 10 in Appendix E contains all experimental settings, including values for the moduli $q$, preprocessing settings, and model training settings.

## 3 Data Preprocessing

Prior work primarily used `BKZ2.0` in the preprocessing/lattice reduction step. While effective, `BKZ2.0` is slow for large values of $n$. We found that for $n = 1024$ matrices it could not finish a single loop in 3 days on an Intel Xeon Gold 6230 CPU, eventually timing out.

**Our preprocessing pipeline.** Our improved preprocessing incorporates a recently developed (at the time of writing) reduction algorithm `Flatter` Ryan & Heninger (2023)[1], which promises similar reduction guarantees to `LLL` with much-reduced compute time, allowing us to preprocess LWE matrices in dimension up to $n = 1024$. We interleave `Flatter` and `BKZ2.0` and switch between them after 3 loops of one results in $\Delta\rho < -0.001$. Following Li et al. (2023b), we run `polish` after each `Flatter` and `BKZ2.0` loop concludes. We initialize our `Flatter` and `BKZ2.0` runs with block size 18 and $\alpha = 0.04$, which provided the best empirical trade-off between time and reduction quality (see Appendix F for additional details), and make reduction parameters stricter as reduction progresses—up to block size 22 for `BKZ2.0` and $\alpha = 0.025$ for `Flatter`.

**Preprocessing performance.** Table 4 records the reduction $\rho$ achieved for each $(n, q)$ pair and the time required, compared with Li et al. (2023b). Recall that $\rho$ measures the reduction in the standard deviation of $\mathbf{A_i}$ relative to its original uniform random distribution; lower is better. Standard deviation strongly correlates with vector norms, but for consistency with Li et al. (2023b) we use standard deviation.

We record reduction time as CPU · hours · matrix, the amount of time it takes our algorithm to reduce one LWE matrix using one CPU. We parallelize our reduction across many CPUs. For $n = 512$, our methods improve reduction time by a factor of 25, and scale easily to $n = 1024$ problems. Overall, we find that `Flatter` improves the *time* (and consequently *resources required*) for preprocessing, but does not improve the overall reduction quality.

---

[1]https://github.com/keeganryan/flatter

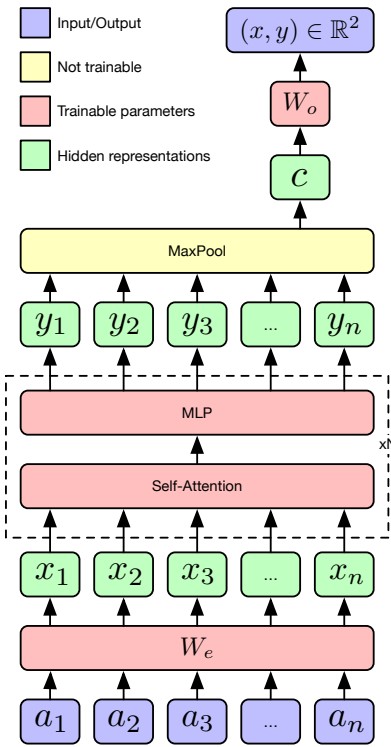

Figure 1: **Encoder-only transformer (§4.1) with angular embedding.** See §4.2 for an explanation of our proposed angular embedding.

**Error penalty** $\omega$**.** We run all reduction experiments with penalty $\omega = 10$. Table 5 demonstrates the tradeoff between reduction quality and reduction error, as measured by $\|\mathbf{R}\|/q$, for $n = 1024$, $\log_2 q = 50$ problems. Empirically, we find that $\|\mathbf{R}\|/q < 0.09$ is sufficiently low to recover secrets from LWE problems with $\mathbf{e} \sim \mathcal{N}(0, 3^2)$.

**Experimental setup.** In practice, we do not preprocess a different set of $4n$ $(\mathbf{a}, b)$ pairs for each secret recovery experiment because preprocessing is so expensive. Instead, we use a single set of preprocessed $\mathbf{Ra}$ rows combined with an arbitrary secret to produce different $(\mathbf{Ra}, \mathbf{R}b)$ pairs for training. We first generate a secret $\mathbf{s}$ and calculate $\mathbf{b} = \mathbf{A} \cdot \mathbf{s} + e$ for the original $4n$ pairs. Then, we apply the many different $\mathbf{R}$ produced by preprocessing to $\mathbf{A}$ and $\mathbf{b}$ to produce many $(\mathbf{Ra}, \mathbf{R}b)$ pairs with reduced norm. This technique enables analyzing attack performance across many dimensions (varying $h$, model parameters, etc.) in a reasonable amount of time. Preprocessing a new dataset for each experiment would make evaluation at scale near-impossible.

# 4 Model Architecture

Previous ML attacks on LWE use a encoder-decoder transformer (Vaswani et al., 2017). A bidirectional encoder processes the input and an auto-regressive decoder generates the output. Integers in both input and output sequences were tokenized as two digits in a large base smaller than $q$. We propose a simpler and faster encoder-only model and introduce an angular embedding for integers modulo $q$.

## 4.1 Encoder-only model

Encoder-decoder models were originally introduced for machine translation, because their outputs can be longer than their inputs. However, they are complex and slow at inference, because the decoder must run once for each output token. For LWE, outputs (one integer) are always shorter than inputs (a vector of $n$

integers). Li et al. (2023b) observe that an encoder-only model, a 4-layer bidirectional transformer based on DeBERTa (He et al., 2020), achieves comparable performance with their encoder-decoder model.

Here, we experiment with simpler encoder-only models without DeBERTa's disentangled attention mechanism with 2 to 8 layers. Outputs are max-pooled across the sequence dimension, decoded by a linear layer for each output digit (Figure 1). We minimize a cross-entropy loss. This simpler architecture improves training speed by 25%.

### 4.2 Angular embedding

Most transformers process sequences of tokens from a fixed vocabulary, encoded in $\mathbb{R}^d$ by a learned embedding. Typical vocabulary sizes vary from 32K in Llama2 (Touvron et al., 2023), to 256K in Jurassic-1 (Lieber et al., 2021). The larger the vocabulary, the more data needed to learn the embeddings. LWE inputs and outputs are integers from $\mathbb{Z}_q$. For $n \geq 512$ and $q \geq 2^{35}$: encoding integers with one token creates a too-large vocabulary. To avoid this, Li et al. (2023a;b) encode integers with two tokens by representing them in base $B = \frac{q}{k}$ with small $k$ and binning the low digit so the overall vocabulary has $< 10\text{K}$ tokens.

This approach has two limitations. First, input sequences are $2n$ tokens long, which slows training as $n$ grows, because transformers' attention mechanism scales quadratically in sequence length. Second, beyond sequence length concerns, vocabulary-based embeddings do not introduce any inductive bias: models must learn how to embed tokens solely from the data. However, LWE is a structured problem (modular arithmetic over $\mathbb{Z}_q$), and prior work has demonstrated that transformers can learn this structure. Liu et al. (2022) showed that transformers trained on modular arithmetic problems learn vocabulary embeddings that mirror the circle-like structure present in $\mathbb{Z}_q$ (e.g. the embedding for 0 is close to that of 1 and $q-1$) late in training, a phenomenon known as "grokking" Power et al. (2022); Gromov (2023).

To address these shortcomings, we introduce an **angular embedding** which strives to better represent the problem's modular structure in embedding space, while encoding integers with only one token. An integer $a \in \mathbb{Z}_q$ is first converted to an angle by the transformation $a \rightarrow 2\pi\frac{a}{q}$, and then to the point $(\sin(2\pi\frac{a}{q}), \cos(2\pi\frac{a}{q})) \in \mathbb{R}^2$. All input integers (in $\mathbb{Z}_q$) are therefore represented as points on the 2-dimensional unit circle, which is then embedded as an ellipse in $\mathbb{R}^d$, via a learned linear projection $W_e$. This improves the sequence length: individual elements of $a$ each only need one token, so $a \in \mathbb{Z}_q^n$ uses $n$ tokens instead of $2n$ tokens as in prior work. Our approach also adds inductive bias: because we *linearly* project from the unit circle in $\mathbb{R}^2$ to the transformer's embedding space $\mathbb{R}^d$, the embeddings for the different elements in $\mathbb{Z}_q$ are arranged in a ellipsis, preserving the circle-like structure of integers mod $q$.

Model outputs, obtained by max-pooling the encoder output sequence, are decoded as points in $\mathbb{R}^2$ by another linear projection $W_o$. The training loss is the $\mathbb{L}_2$ distance between the model prediction and the point representing $b$ on the unit circle.

### 4.3 Experiments

Here, we compare our new architecture with previous work, and assess its performance on larger instances of LWE ($n = 768$ and $1024$). All comparisons with prior work are performed on $n = 512$ and $\log_2 q = 41$ for binary and ternary secrets, using the same pre-processing techniques as Li et al. (2023b).

**Encoder-only models vs prior designs.** In Table 6, we compare our encoder-only model (with and without the angular embedding) with the encoder-decoder and the DeBERTa models from Li et al. (2023b). The encoder-only and encoder-decoder models are trained for 72 hours on one 32GB V100 GPU.

The DeBERTa model, which requires more computing resources, is trained for 72 hours on four 32GB V100 GPUs. Our encoder-only model, using the same vocabulary embedding as prior work, processes samples 25% faster than the encoder-decoder architecture and is 3× faster than the DeBERTa architecture. With the angular embedding, training is 2.4× faster, because input sequences are half as long, so attention calculations are accelerated. Our models also outperform prior designs on secret recovery: previous models recover binary secrets with Hamming weight 63 and ternary secrets with Hamming weight 60. Encoder-only models with an

Table 6: **Best recovery results for binary and ternary secrets on various model architectures** ($n = 512$, $\log_2 q = 41$). Encoder-Decoder and DeBERTa models and recovery results are from Li et al. (2023b); we benchmark DeBERTA samples/sec on our hardware. Encoder (Vocab.) uses prior work's vocabulary embedding. Encoder (Angular) is presented in Section 4.2.

| Architecture | Samples/ Sec | Largest Binary $h$ | Largest Ternary $h$ |
|---|---|---|---|
| Encoder-Decoder | 200 | 63 | 58 |
| Encoder (DeBERTa) | 83 | 63 | 60 |
| Encoder (Vocab.) | 256 | 63 | **66** |
| Encoder (Angular) | 610 | **66** | **66** |

Figure 2: **Encoder-only (angular embedding) performance for varying # of layers and embedding dimensions** ($n = 512, \log_2 q = 41$, binary $\chi_{\mathbf{s}}$). Samples per second quantifies training speed; "Recovered" = % recovered out of 100 secrets with $h$ from 49-67; "Hours"= mean hours to recovery.

| Layers | Emb. Dim. | Params | Samples/S | Recovered | Hours |
|---|---|---|---|---|---|
| 2 | 128 | 1.3M | **2560** | 23% | **18.9** |
| 4 | 256 | 4.1M | 1114 | 22% | 19.6 |
| 4 | 512 | 14.6M | 700 | **25%** | 26.2 |
| 6 | 512 | 20.9M | 465 | **25%** | 28.1 |
| 8 | 512 | 27.2M | 356 | 24% | 30.3 |

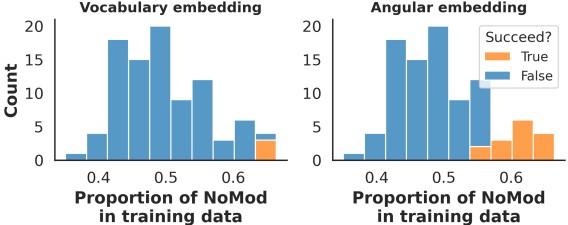

Figure 3: **Count of # successes (orange) and failures (blue) for various NoMod for vocabulary-based vs. angular embedding schemes.** $n = 512, \log_2 q = 41, h = 57\text{-}67$.

angular embedding recover binary and ternary secrets with Hamming weights up to 66. §G.1 in the Appendix gives detailed results.

**Impact of model size.** Table 2 compares encoder-only models of different sizes (using angular embeddings). All models are run for up to 72 hours on one 32GB V100 GPU on $n = 512, \log_2 q = 41$. We observe that larger models yield little benefit in terms of secret recovery rate, and small models are significantly faster (both in terms of training and recovery). Additional results are in Appendix G.2. We use 4 layers with embedding dimension 512 for later experiments because it recovers the most secrets (25%) the fastest.

**Embedding ablation.** Next, we compare our new angular embedding scheme to the vocabulary embeddings. The better embedding should recover both more secrets and more difficult ones (as measured by Hamming weight and **NoMod**; see §2.2 for a description of **NoMod**). To measure this, we run attacks on identical datasets, $n = 512, \log_2 q = 41, h = 57\text{-}67$ with 10 unique secrets per $h$. One set of models uses the angular embedding, while the other uses the vocabulary embedding from Li et al. (2023b).

To check if angular embedding outperforms vocabulary embedding, we measure the attacked secrets' **NoMod**. We expect the better embedding to recover secrets with lower **NoMod** and higher Hamming weights (e.g. harder secrets). As Figure 3 and Table 6 demonstrate, this is indeed the case. Angular embeddings recover secrets with **NoMod**= 56 vs. 63 for vocabulary embedding (see Table 29 in Appendix I for raw numbers). Furthermore, angular embedding models recover more secrets than those with vocabulary embeddings (16 vs. 2) and succeed on higher $h$ (66 vs. 63). We conclude that an angular embedding is superior to a vocabulary embedding because it recovers harder secrets.

**Scaling $n$.** Finally, we use our proposed architecture improvements to scale $n$. The long input sequence length in prior work made scaling attacks to $n \geq 512$ difficult, due to both memory footprint and slow model processing speed. In contrast, our more efficient model and angular embedding scheme (§4.1,§4.2) enable us to attack $n = 768$ and $n = 1024$ secrets. Table 7 shows that we can recover up to $h = 9$ for both $n = 768$ and $n = 1024$ settings, with $< 24$ hours of training on a single 32GB V100 GPU. In §5.1 we show recovery of $h = 13$ for $n = 1024$ using a more sample-efficient training strategy. We run identical experiments using prior work's encoder-decoder model, but fail to recover any secrets with $n > 512$ in the same computational budget.

Table 7: **Secret recovery time for larger dimensions $n$ with encoder-only model with 4 layers, embedding dimension of 512 and angular embedding.** We only report the training hours (on one V100 GPU) for *successful* secret recoveries, out of 10 secrets per $h$ value.

| $n$ | $\log_2 q$ | Samples/ Sec | Hours to recovery for different $h$ values | | | |
|-----|-----------|------|------------|------------|------------|------------|
| | | | $h = 5$ | $h = 7$ | $h = 9$ | $h = 11$ |
| 768 | 35 | 355 | 3.1, 18.6, 18.9 | 9.1, 21.6, 24.9 | 15.9, 27.7 | - |
| 1024 | 50 | 256 | 1.6, 6.2, 7.6, 8.8, 34.0, 41.4, 43.7 | 4.6, 7.4, 13.5, 16.7 | 21.3 | - |

Table 8: **# Training samples needed to recover secrets without pre-training.** ($n = 512, \log_2 q = 41$, binary secrets). We report # of secrets recovered among $h = 30\text{-}45$ (10 secrets for each $h$), the highest $h$ recovered, and the average attack time among secrets recovered with 1M, 3M and 4M training samples.

| Training samples | Total # | Best $h$ | Mean Hours |
|-----|-----|-----|-----|
| 300K | 1 | 32 | 30.3 |
| 1M | 18 | 44 | $28.0 \pm 11.6$ |
| 3M | 21 | 44 | $26.5 \pm 10.5$ |
| 4M | 22 | 44 | $25.3 \pm 8.9$ |

Our proposed model improvements lead to the first successful ML attack that scales to real-world values of $n$: proposed real-world use cases for LWE-based cryptosystems recommend using dimensions $n = 768$ and $n = 1024$ Avanzi et al. (2021); Albrecht et al. (2021); Curtis & Player (2019), although they also recommend smaller $q$ and harder secret distributions.

## 5 Training Methods

The final limitation we address is the 4 million preprocessed LWE samples for model training. Recall that each training sample is a row of a reduced LWE matrix $\mathbf{RA} \in \mathbb{Z}_q^{m \times n}$, so producing 4 million training samples requires reducing $\approx \frac{4,000,000}{m+n}$ LWE matrices. Even with the preprocessing improvements highlighted in §3, for $n = 1024$, this means preprocessing between 2000 and 4500 matrices at the cost of 26 hours per CPU per matrix. [2] To further reduce total attack time, we propose training with fewer samples and pre-training models.

### 5.1 Training with Fewer Samples

We first consider simply reducing training dataset size and seeing if the attack succeeds. Li et al. always use 4M training examples. To test if this many is needed for secret recovery, we subsample datasets of size $N = [100K, 300K, 1M, 3M]$ from the original 4M examples preprocessed via the techniques in §3. We train models to attack LWE problems $n = 512, \log_2 q = 41$ with binary secrets and $h = 30\text{-}45$. Each attack is given 3 days on a single V100 32GB GPU.

Table 8 shows that our attack still succeeds, even when only 300K samples are used. We recover approximately the same number of secrets with 1M samples as with 4M, and both settings achieve the same best $h$. Using 1M rather than 4M training samples reduces our preprocessing time by 75%. We run similar experiments for $n = 768$ and $n = 1024$, and find that we can recover up to $h = 13$ secrets for $n = 1024$ with only 1M training samples. Those results are in Tables 17 to 20 in Appendix H.1.

### 5.2 Model Pre-Training

To further improve sample efficiency, we introduce the first use of pre-training in LWE. Pre-training models has improved sample efficiency in language (Devlin et al., 2019; Brown et al., 2020) and vision (Kolesnikov et al., 2020); we hypothesize similar improvements for LWE. We frame secret recovery as a downstream

---

[2]We bound the number of reduced matrices needed because some rows of reduction matrices $\mathbf{R}$ are $\mathbf{0}$, discarded after preprocessing. Between $m$ and $n + m$ nonzero rows of $\mathbf{R}$ are kept, so we must reduce between $\frac{4,000,000}{m+n}$ and $\frac{4,000,000}{m}$ matrices.

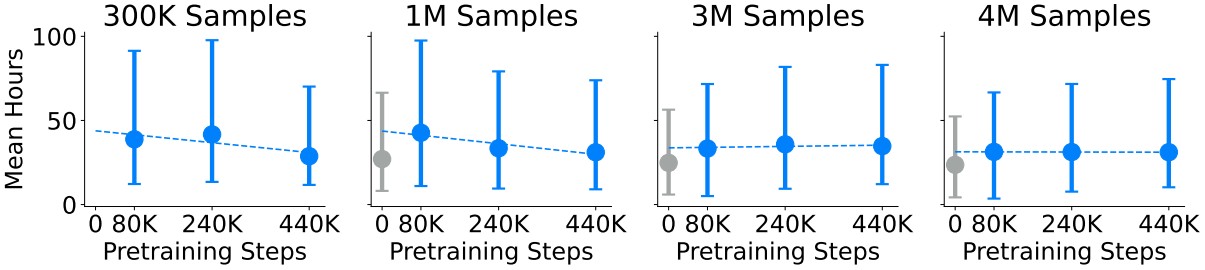

Figure 4: **How pre-training affects mean hours to secret recovery for different training dataset sizes.** ($n = 512$, $\log_2 q = 41$, binary secrets).

task and pre-train a transformer to improve sample efficiency when recovering new secrets, further reducing preprocessing costs.

Formally, an attacker would like to pre-train a model parameterized by $\theta$ on samples $\{(\mathbf{a}', b')\}$ such that the pre-trained parameters $\theta^*$ are a better-than-random initialization for recovering a secret from new samples $\{(\mathbf{a}, b)\}$. Although pre-training to get $\theta^*$ may require significant compute, $\theta^*$ can initialize models for many different secret recoveries, amortizing the initial cost.

**Pre-training setup.** First, we generate and reduce a new dataset of 4 million $\mathbf{Ra}'$ samples with which we pre-train a model. The $\theta^*$-initialized model will then train on the $(\mathbf{Ra}, \mathbf{R}b)$ samples used in §5.1, leading to a fair comparison between a randomly-initialized model and a $\theta^*$-initialized model. Pre-training on the true attack dataset is unfair and unrealistic, since we assume the attacker will train $\theta^*$ before acquiring the real LWE samples they wish to attack.

The pre-trained weights $\theta^*$ should be a good initialization for recovering many different secrets. Thus, we use many different secrets with the 4M rows $\mathbf{Ra}'$. In typical recovery, we have 4M rows $\mathbf{Ra}$ and 4M targets $\mathbf{R}b$. In the pre-training setting, however, we generate 150 different secrets so that each $\mathbf{Ra}'$ has 150 different possible targets. So the model can distinguish targets, we introduce 150 special vocabulary tokens $t_{s_i}$, one for each secret $s_i$. We concatenate the appropriate token $t_{s_i}$ to row $\mathbf{Ra}'$ paired with $\mathbf{R}b' = \mathbf{Ra}' \cdot \mathbf{s_i} + \mathbf{e}$. Thus, from 4M rows $\mathbf{Ra}'$, we produce 600M triplets $(\mathbf{Ra}', t_{s_i}, \mathbf{R}b')$. The model learns to predict $\mathbf{R}b'$ from a row $\mathbf{Ra}'$ and an integer token $t_{s_i}$.[3]

We hypothesize that including many $\mathbf{Ra}', \mathbf{R}b'$ pairs produced from different secrets will induce strong generalization capabilities. When we train the $\theta^*$-initialized model on new data with an unseen secret $\mathbf{s}$, we indicate that there is a new secret by adding a new token $t_0$ to the model vocabulary that serves the same function as $t_{s_i}$ above. We randomly initialize the new token embedding, but initialize the remaining model parameters with those of $\theta^*$. Then we train and extract secrets as in §2.2.

**Experiments.** For pre-training data, we use 4M $\mathbf{Ra}'$ rows reduced to $\rho = 0.41$, generated from a new set of $4n$ LWE $(\mathbf{a}, b)$ samples. We use binary and ternary secrets with Hamming weights from 30 to 45, with 5 different secrets for each weight, for a total of 150 secrets and 600M $(\mathbf{Ra}', t_{s_i}, \mathbf{R}b')$ triplets for pre-training. We pre-train an encoder-only transformer with angular embeddings for 3 days on 8x 32GB V100 GPUs with a global batch size of 1200. The model sees 528M total examples, less than one epoch. We do not run the distinguisher step during pre-training because we are not interested in recovering these secrets.

We use the weights $\theta^*$ as the initialization and repeat §5.1's experiments. We use three different checkpoints from pre-training as $\theta^*$ to evaluate the effect of different pre-training amounts: 80K, 240K and 440K pre-training steps.

**Results.** Figure 5 demonstrates that pre-training improves sample efficiency during secret recovery. We record the minimum number of samples required to recover each secret for each checkpoint. Then we average these minimums among secrets recovered by all checkpoints (including the randomly initialized model) to fairly compare them. We find that 80K steps improves sample efficiency, dropping from 1.7M to 409K mean samples required. However, further pre-training does not further improve sample efficiency.

---

[3]$t_{s_i}$ is embedded using a learned vocabulary.

Recall that using fewer samples harms recovery speed (see Table 8). Figure 4 shows trends for recovery speed for pre-trained and randomly initialized models for different numbers of training samples. We find that any pre-training slows down recovery, but further pre-training might minimize this slowdown. Appendix H has complete results.

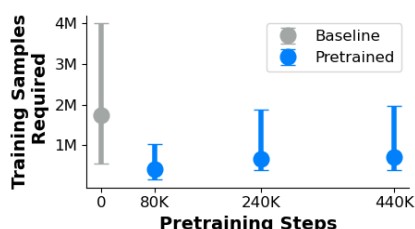

Figure 5: **Mean minimum number of samples needed to recover binary secrets as a function of # pre-training steps.** ($n = 512$, $\log_2 q = 41$, binary secrets).

## 6 Comparison to Other Attacks

It is difficult to compare against other attacks because few of the known LWE attacks have been implemented at scale and have code available, and no prior work has demonstrated successful LWE attack performance for the (large) dimensions we consider. One unpublished work Ducas et al. (2023) provides code and claims to outperform Salsa Verde Li et al. (2023b), but in fact does not, and does not even recover secrets in dimension $n = 512$. We compare against this publicly available code[4] and find that it does not recover any secrets for the parameters we attack.

In addition, we also follow common practice in the lattice community and compare against the LWE Estimator Albrecht et al. (2015)[5], a public Github repository commonly used to provide theoretical estimates of LWE attack performance. However, the authors note that it has not been peer reviewed NIST (2023). Furthermore, it is inaccurate in several instances Chen et al. (2020); Cheon et al. (2019); Ducas & Pulles (2023) and does not actually implement attacks.

We present concrete results using code from Ducas et al. (2023) in Table 9, and the Estimator's predicted best attack for our parameters in Table 1. Ducas et al. (2023)'s code ran for weeks on our LWE parameter sets, but never succeeded. Thus, we report "-" for that attack. In the estimator results, the ROP metric refers loosely to the number of operations needed to run the attack, which does not easily translate to time measurements, making it a crude estimate for runtime. For example, for ($\mathbf{Z}/q\mathbf{Z}$) ring operations, one should multiply by the cost of a multiplication modulo $q$, naively $(\log q)^2$, or for ciphertext operations one should multiply by the cost of polynomial multiplications modulo $q$, etc. We omit space requirements which may make implementation of some of the theoretical attacks impossible for these large dimensions.

Table 9: **Comparison to other claimed attacks.** Estimates from Albrecht et al. (2015) given in terms of ROP = estimated number of required operations for attack. Space requirements not included here.

| **LWE setting** $(n, q, h)$ | **Estimator Results Albrecht et al. (2015)** Attack Name | ROP | **Ducas et al.** 2023 Time (sec) |
|---|---|---|---|
| $(512, 41, 44)$ | Bounded Distance Decoding (BDD) Liu & Nguyen (2013) | $2^{42.8}$ | - |
| $(768, 35, 9)$ | BDD Meet-in-the-middle Hybrid Howgrave-Graham (2007) | $2^{46.1}$ | - |
| $(1024, 50, 13)$ | BDD Meet-in-the-middle Hybrid Howgrave-Graham (2007) | $2^{48.1}$ | - |

## 7 Discussion & Future Work

Our contributions are spread across multiple fronts: faster preprocessing ($25\times$ fewer CPU hours), simpler architecture ($25\%$ more samples/sec), better token embeddings ($2.4\times$ faster training) and the first use of pre-training for LWE ($10\times$ fewer samples). Our efforts lead to **$250\times$ fewer CPU hours spent preprocessing** and **$3\times$ more samples/sec** for for $n = 512, \log_2 q = 41$ LWE problems, and lead to the **first ML attack on LWE for n = 768 and n = 1024**. Of note is that our contributions compose well; rarely in machine learning research are optimizations across multiple fronts truly independent. A hypothetical $2\times$ speedup via data filtering and $3\times$ speedup from an improved optimization algorithm is unlikely to combine to a $6\times$ speedup. In contrast, our optimizations do compose well.

---

[4] https://github.com/lducas/leaky-LWE-Estimator/tree/human-LWE/human-LWE

[5] https://github.com/malb/lattice-estimator, commit 00ec72ce

Although we have made substantial progress in pushing the boundaries of machine learning-based attacks on LWE, much future work remains in both building on this pre-training work and improving models' capacity to learn modular arithmetic.

## 8 Limitations

Our work still does not attack concrete proposed LWE parameters (for example, the HE Standard Albrecht et al. (2021) proposes $n = 1024, \log_2 q = 29$ with non-sparse secrets for an equivalent 128-bit security level). We are primarily limited in two directions. First, wow effectively lattice reduction methods reduce samples. Our work integrates and leverages the latest developments in lattice reduction to improve the number of samples with scalar products $\mathbf{a} \cdot \mathbf{s}$ that lie in $(-\frac{q}{2}, \frac{q}{2})$; this property is referred to as **NoMod** in §2.2 and prior work (Li et al., 2023b). Second, we are limited by the ability of our neural network (currently a bidirectional encoder) to learn from samples that do *not* satisfy this **NoMod** property. While prior work has demonstrated that transformers can learn modular arithmetic for $n = 1$ (Liu et al., 2022), current architectures struggle with larger values of $n$.

Furthermore, while pre-training improves sample efficiency, it does not lead to improvements when recovering secrets with 3M or more reduced samples. While disappointing, this is not surprising: the "pre-train/fine-tune" paradigm primarily improves over "train-from-scratch" in small-data regimes. Future work, instead of improving recovery speed with millions of samples, should focus on further improving sample efficiency to 100K or fewer.

### Acknowledgments

We thank Mark Tygert for his always insightful comments and Mohamed Malhou for running experiments.

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

## A  Appendix

We provide details, additional experimental results, and analysis ommitted in the main text:

1. Appendix B: Broader impacts
2. Appendix C: Related work
3. Appendix D: Future work
4. Appendix E: Attack parameters
5. Appendix F: Lattice reduction background
6. Appendix G: Additional model results (Section 4)
7. Appendix H: Additional training results (Section 5)
8. Appendix I: Further **NoMod** analysis

## B  Broader Impacts

The main ethical concern related to this work is the possibility of our attack compromising currently-deployed PQC system. However, at present, our proposed attack does not threaten current standardized systems. If our attack scales to higher $h$ and lower $q$ settings, then its impact is significant, as it would necessitate changing PQC encryption standards.

## C  Related Work

**Using machine learning for cryptanalysis.** An increasing number of cryptanalytic attacks in recent years have incorporated ML models. Often, models are used to strengthen existing cryptanalysis approaches, such as side channel or differential analysis Chen & Yu (2021). Of particular interest is recent work that successfully used ML algorithms to aid side-channel analysis of Kyber, a NIST-standardized PQC method Dubrova et al. (2022). Other ML-based cryptanalysis schemes train models on plaintext/ciphertext pairs or similar data, enabling direct recovery of cryptographic secrets. Such approaches have been studied against a variety of cryptosystems, including hash functions Goncharov (2019), block ciphers Gohr (2019); Benamira et al. (2021); Chen & Yu (2021); Alani (2012); So (2020); Kimura et al. (2021); Baek & Kim (2020), and substitution ciphers Ahmadzadeh et al. (2021); Srivastava & Bhatia (2018); Aldarrab & May (2020); Greydanus (2017). The three ML-based LWE attacks upon which this work builds, Salsa Wenger et al. (2022), Picante Li et al. (2023a), and Verde Li et al. (2023b), also take this approach.

**AI for Math.** The use of neural networks for arithmetic was first considered by Siu & Roychowdhury (1992), and recurrent networks by Zaremba et al. (2015), Kalchbrenner et al. (2015) and Kaiser & Sutskever (2015). Transformers have been used to solve problems in symbolic and numerical mathematics, integration(Lample & Charton, 2020), linear algebra (Charton, 2022), arithmetic (Charton, 2024) and theorem proving (Polu & Sutskever, 2020). With the advent of large language models, recent research has focused on training or fine-tuning language models on math word problems: problems of mathematics expressed in natural language (Meng & Rumshisky, 2019; Griffith & Kalita, 2021; Lee et al., 2023). The limitations of these approaches were discussed by Nogueira et al. (2021) and Dziri et al. (2023). Modular arithmetic was first considered by Power et al. (2022) and Wenger et al. (2022). Its difficulty was discussed by Palamas (2017) and Gromov (2023).

## D  Future Work

**Pre-training.** Pre-training improves sample efficiency after 80K steps; however, further improvements to pre-training should be explored. First, our experiments used just one set of 4M **RA** combined with multiple secrets. To encourage generalization to new **RA**, pre-training data should include different original **A**s. Second, our transformer only has 14.1M parameters, and may be too small to benefit from pre-training. Third, pre-training data does not have to come from a sniffed set of $4n$ $(\mathbf{a}, b)$ samples. Rather than use expensive preprocessed data, we could simulate reduction and generate random rows synthetically that look like **RA**.

**ML for modular arithmetic. NoMod** experiments consistently show that more training data which does not wrap around the modulus leads to more successful secret recovery (see Figure 3 and Li et al. (2023b)). This explains why a smaller modulus $q$ is harder for ML approaches to attack, and indicates that models are not yet learning modular arithmetic (Wenger et al., 2022). Further progress on models learning modular arithmetic problems will likely help to achieve secret recovery for smaller $q$ and larger $h$.

# E    Parameters for our attack

For training all models, we use a learning rate of $10^{-6}$, weight decay of 0.001, 3000 warmup steps, and an embedding dimension of 512. We use 4 encoder layers, and 8 attention heads for all encoder-only model experiments, except the architecture ablation in Figure 2. We use the following primes: $q = 2199023255531$ for $n = 512$, $q = 34088624597$ for $n = 768$, and $q = 607817174438671$ for $n = 1024$. The rest of the parameters used for the experiments are in Table 10.

Table 10: **LWE, preprocessing, and training parameters.** For the adaptive increase of preprocessing parameters, we start with block size $\beta_1$, flatter $\alpha_1$, and LLL-delta $\delta_1$ and upgrade to $\beta_2$, $\alpha_2$, and $\delta_2$ at a later stage. Parameters base $B$ and bucket size are used to tokenize the numbers for transformer training.

| LWE parameters | | | Preprocessing settings | | | | | | Model training settings | | |
|---|---|---|---|---|---|---|---|---|---|---|---|
| $n$ | $\log_2 q$ | $h$ | $\beta_1$ | $\beta_2$ | $\alpha_1$ | $\alpha_2$ | $\delta_1$ | $\delta_2$ | Base $B$ | Bucket size | Batch size |
| 512 | 41 | $\leq 70$ | 18 | 22 | 0.04 | 0.025 | 0.96 | 0.9 | 137438953471 | 134217728 | 256 |
| 768 | 35 | $\leq 11$ | 18 | 22 | 0.04 | 0.025 | 0.96 | 0.9 | 1893812477 | 378762 | 128 |
| 1024 | 50 | $\leq 11$ | 18 | 22 | 0.04 | 0.025 | 0.96 | 0.9 | 25325715601611 | 5065143120 | 128 |

# F    Additional background on lattice reduction algorithms

Lattice reduction algorithms reduce the length of lattice vectors, and are a building block in most known attacks on lattice-based cryptosystems. If a short enough vector can be found, it can be used to recover LWE secrets via the dual, decoding, or uSVP attack Micciancio & Voulgaris (2010); Albrecht et al. (2017; 2021). The original lattice reduction algorithm, `LLL` Lenstra et al. (1982), runs in time polynomial in the dimension of the lattice, but is only guaranteed to find an exponentially bad approximation to the shortest vector. In other words the *quality* of the reduction is poor. `LLL` iterates through the ordered basis vectors of a lattice, projecting vectors onto each other pairwise and swapping vectors until a shorter, re-ordered, nearly orthogonal basis is returned.

To improve the quality of the reduction, the `BKZ` Schnorr (1987) algorithm generalizes `LLL` by projecting basis vectors onto $k-1$-dimensional subspaces, where $k < n$ is the "blocksize". (LLL has blocksize 2.) As $k$ approaches $n$, the quality of the reduced basis improves, but the running time is exponential in $k$, so is not practical for large block size. Experiments in Chen et al. (2020) with running `BKZ` to attack LWE instances found that block size $k \geq 60$ and $n > 100$ was infeasible in practice. Both `BKZ` and `LLL` are implemented in the *fplll* library The FPLLL development team (2023), along with an improved version of `BKZ`: `BKZ2.0` Chen & Nguyen (2011). Charton et al. proposed an alternative lattice reduction algorithm similar to `LLL`. In practice we use it as a `polishing` step after each `BKZ` loop concludes. It "polishes" by iteratively orthogonalizing the vectors, provably decreasing norms with each run.

A newer alternative to `LLL` is `flatter` Ryan & Heninger (2023), which provides reduction guarantees analogous to `LLL` but runs faster due to better precision management. `flatter` runs on sublattices, and leverages clever techniques for reducing numerical precision as the reduction proceeds, enabling it to run much faster than other reduction implementations. Experiments in the original paper show `flatter` running orders of magnitude faster than other methods on high-dimensional ($n \geq 1024$) lattice problems. The implementation of `flatter`[6] has a few tunable parameters, notably $\alpha$, which characterizes the strength of the desired reduction in terms of lattice "drop", a bespoke method developed by Ryan & Heninger (2023) that mimics the Lovász condition in traditional LLL Lenstra et al. (1982). In our runs of `flatter`, we set $\alpha = 0.04$ initially and

---

[6]https://github.com/keeganryan/flatter

decrease it to $\alpha = 0.025$ after the lattice is somewhat reduced, following the adaptive reduction approach of Li et al. (2023b).

## G   Additional Results for §4

### G.1   Architecture Comparison

Tables 11, 12, (for binary secrets) and Tables 13, and 14 (for ternary secrets) expand on the results in Table 6 by showing how three different model architectures perform on binary and ternary secrets with different Hamming weights. We see that the encoder-only model architecture with angular embedding improves secret recovery compared to the encoder-decoder model and the encoder-only model with two-token embedding. Notably, the encoder-only model with angular embedding is able to recover secrets up to $h = 66$ for both binary and ternary secrets, which is a big improvement compared to previous work.

Table 11: **Secret recovery (# successful recoveries/# attack attempts) with various model architectures for $h = 57$ to $h = 66$ ($n = 512, \log_2 q = 41$, binary secrets).**

| Architecture | $h$ | | | | | | | | | |
| --- | --- | --- | --- | --- | --- | --- | --- | --- | --- | --- |
| | 57 | 58 | 59 | 60 | 61 | 62 | 63 | 64 | 65 | 66 |
| Encoder-Decoder | 1/10 | | 1/10 | | | | 1/10 | | | |
| Encoder (Vocabulary) | 0/10 | 0/7 | 2/10 | 0/10 | 0/8 | 0/10 | 1/10 | 0/9 | 0/8 | 0/8 |
| Encoder (Angular) | **2/10** | 0/7 | **3/10** | **2/10** | **1/8** | **1/10** | **2/10** | **1/9** | **1/8** | **2/8** |

Table 12: **Average time to recovery (hours) for successful recoveries with various model architectures for $h = 57$ to $h = 66$ ($n = 512, \log_2 q = 41$, binary secrets).**

| Architecture | $h$ | | | | | | | | | |
| --- | --- | --- | --- | --- | --- | --- | --- | --- | --- | --- |
| | 57 | 58 | 59 | 60 | 61 | 62 | 63 | 64 | 65 | 66 |
| Encoder-Decoder | 10.0 | - | 20.0 | - | - | - | 17.5 | - | - | - |
| Encoder (Vocabulary) | - | - | **19.8** | **-** | - | **-** | **22.0** | - | **-** | **-** |
| Encoder (Angular) | 33.0 | - | 37.0 | 33.2 | 29.1 | 26.7 | 29.8 | 57.1 | 31.6 | 28.8 |

Table 13: **Secret recovery (# successful recoveries/# attack attempts) with various model architectures for $h = 57$ to $h = 66$ ($n = 512, \log_2 q = 41$, ternary secrets).**

| Architecture | $h$ | | | | | | | | | |
| --- | --- | --- | --- | --- | --- | --- | --- | --- | --- | --- |
| | 57 | 58 | 59 | 60 | 61 | 62 | 63 | 64 | 65 | 66 |
| Encoder-Decoder | - | 1/10 | - | - | - | - | - | - | - | - |
| Encoder (Vocabulary) | 0/9 | 1/8 | **0/9** | **0/9** | 1/10 | **0/8** | **0/7** | 0/7 | **0/10** | 1/9 |
| Encoder (Angular) | 0/9 | 1/8 | 0/9 | 0/9 | 1/10 | 0/8 | 0/7 | 0/7 | 0/10 | 1/9 |

### G.2   Architecture Ablation

Here, we present additional results from the architecture ablation experiments summarized in Table 2. The results in Table 15 and Table 16 show the number of successful recoveries and average time to recovery with varying architectures across different Hamming weights. We see that increasing transformer depth (number of layers) tends to improve recovery but also increases average recovery time. Increasing the embedding dimension from 256 to 512 with 4 layers improves secret recovery. Thus, we choose 4 layers with a hidden dimension of 512 as it recovers the most secrets (25%) the fastest (26.2 mean hours).

Table 14: **Average time to recovery (hours) for successful recoveries with various model architectures for $h = 57$ to $h = 66$** ($n = 512, \log_2 q = 41$, ternary secrets).

| Architecture | | | | | $h$ | | | | | |
|---|---|---|---|---|---|---|---|---|---|---|
| | 57 | 58 | 59 | 60 | 61 | 62 | 63 | 64 | 65 | 66 |
| Encoder-Decoder | - | 27.5 | - | - | - | - | - | - | - | - |
| Encoder (Vocabulary) | - | **24.8** | - | - | **28.2** | - | - | - | - | **34.4** |
| Encoder (Angular) | - | 57.6 | - | - | 47.4 | - | - | - | - | 70.3 |

Table 15: **Effect of different transformer depths (# of layers) and widths (embedding dimension) on secret recovery (# successful recoveries/# attack attempts) with encoder-only model** ($n = 512$, $\log_2 q = 41$, binary secrets).

| Layers | Emb Dim | | | | | $h$ | | | | | |
|---|---|---|---|---|---|---|---|---|---|---|---|
| | | 49 | 51 | 53 | 55 | 57 | 59 | 61 | 63 | 65 | 67 |
| 2 | 128 | 2/9 | 4/10 | 5/10 | 3/9 | 2/10 | 2/8 | 1/9 | **3/10** | 1/9 | 0/10 |
| 4 | 256 | 2/9 | 4/10 | 5/10 | 3/9 | 2/10 | 2/8 | 1/9 | 2/10 | 1/9 | 0/10 |
| 4 | 512 | **3/9** | 4/10 | 5/10 | **4/9** | 2/10 | 2/8 | 1/9 | **3/10** | 1/9 | 0/10 |
| 6 | 512 | **3/9** | 4/10 | 5/10 | 3/9 | 2/10 | 2/8 | **2/9** | **3/10** | 1/9 | 0/10 |
| 8 | 512 | **3/9** | 4/10 | 5/10 | **4/9** | 2/10 | 2/8 | 1/9 | 2/10 | 1/9 | 0/10 |

# H  Additional Results for §5

## H.1  Training with Fewer Samples

Here, we present additional results from scaling $n$ to 768 and 1024 (without pre-training) as summarized in Table 7. Tables 17 and 18 show the results for the $n = 768, \log_2 q = 35$ case with binary secrets. Similarly, Tables 19 and 20 show the results for the $n = 1024, \log_2 q = 50$ case with binary secrets.

## H.2  Model Pre-Training

In this section, we expand upon the pre-training results summarized in Figures 5 and 4. For each pre-training checkpoint, we measure number of successful recoveries out of 10 trials per $h$ and the average time in hours to successful secret recovery for $h = 30$ to $h = 45$. We also vary the number of samples from 100K to 4M to see which setup is most sample efficient. In all of these experiments, $n = 512$ and $q$ are fixed, $\log_2 q = 41$, with binary secrets. The results are presented as follows: no pre-training baseline (Tables 21 and 22), 80K steps pre-training (Tables 23 and 24), 240K steps pre-training (Tables 25 and 26), and 440K steps pre-training (Tables 27 and 28).

Based on these results, we conclude that some pre-training helps to recover secrets with less samples, but more pre-training is not necessarily better. We also see that more pre-training increases the average time to successful secret recovery.

Table 16: **Effect of different transformer depths (# of layers) and widths (embedding dimension) on average time to recovery (hours) with encoder-only model** ($n = 512$, $\log_2 q = 41$, binary secrets).

| Layers | Emb Dim | $h$ | | | | | | | | | |
|---|---|---|---|---|---|---|---|---|---|---|---|
| | | 49 | 51 | 53 | 55 | 57 | 59 | 61 | 63 | 65 | 67 |
| 2 | 128 | 8.7 | **10.0** | **8.3** | 14.0 | **11.0** | **5.5** | 11.5 | **17.9** | **18.9** | - |
| 4 | 256 | 30.9 | 11.0 | 12.3 | 30.3 | 39.9 | 10.4 | 20.6 | 13.3 | 19.6 | - |
| 4 | 512 | **27.2** | 15.4 | 18.2 | **35.2** | 35.4 | 17.3 | 24.3 | **32.2** | 26.2 | - |
| 6 | 512 | **44.1** | 18.8 | 20.7 | 34.2 | 30.5 | 19.4 | **47.9** | **32.1** | 28.1 | - |
| 8 | 512 | **46.4** | 22.5 | 24.0 | **44.6** | 34.6 | 23.8 | 28.0 | 29.0 | 30.3 | - |

Table 17: **Successful secret recoveries out of 10 trials per $h$ with varying amounts of training data** ($n = 768$, $\log_2 q = 35$, binary secrets).

| # Samples | $h$ | | | |
|---|---|---|---|---|
| | 5 | 7 | 9 | 11 |
| 100K | 1 | - | - | - |
| 300K | 1 | 1 | - | - |
| 1M | 4 | 3 | 2 | - |
| 3M | 2 | 3 | 1 | - |
| 4M | 3 | 3 | 2 | - |

Table 18: **Average time (hours) to successful secret recoveries with varying amounts of training data** ($n = 768$, $\log_2 q = 35$, binary secrets).

| # Samples | $h$ | | | |
|---|---|---|---|---|
| | 5 | 7 | 9 | 11 |
| 100K | 1.4 | - | - | - |
| 300K | 0.7 | 12.3 | - | - |
| 1M | 27.9 | 24.5 | 39.2 | - |
| 3M | 17.8 | 13.0 | 44.2 | - |
| 4M | 13.5 | 18.5 | 21.8 | - |

Table 19: **Successful secret recoveries out of 10 trials per $h$ with varying amounts of training data** ($n = 1024$, $\log_2 q = 50$, binary secrets).

| # Samples | $h$ | | | | | |
|---|---|---|---|---|---|---|
| | 5 | 7 | 9 | 11 | 13 | 15 |
| 100K | 3 | - | - | - | - | - |
| 300K | 6 | 3 | - | - | - | - |
| 1M | 6 | 4 | 1 | - | 1 | - |
| 3M | 6 | 4 | 1 | 1 | - | - |
| 4M | 7 | 5 | 1 | - | - | - |

Table 20: **Average time (hours) to successful secret recoveries with varying amounts of training data** ($n = 1024$, $\log_2 q = 50$, binary secrets).

| # Samples | $h$ | | | | | |
|---|---|---|---|---|---|---|
| | 5 | 7 | 9 | 11 | 13 | 15 |
| 100K | 11.6 | - | - | - | - | - |
| 300K | 14.8 | 17.7 | - | - | - | - |
| 1M | 15.7 | 29.2 | 36.1 | - | 47.4 | - |
| 3M | 14.8 | 14.0 | 29.6 | 39.8 | - | - |
| 4M | 19.9 | 10.6 | 21.3 | - | - | - |

Table 21: **Successful secret recoveries out of 10 trials per $h$ with no model pre-training** ($n = 512, \log_2 q = 41$, binary secrets).

| # Samples | $h$ | | | | | | | | | | | | | | | |
|---|---|---|---|---|---|---|---|---|---|---|---|---|---|---|---|---|
| | 30 | 31 | 32 | 33 | 34 | 35 | 36 | 37 | 38 | 39 | 40 | 41 | 42 | 43 | 44 | 45 |
| 100K | - | - | - | - | - | - | - | - | - | - | - | - | - | - | - | - |
| 300K | - | - | 1 | - | - | - | - | - | - | - | - | - | - | - | - | - |
| 1M | - | - | 3 | 2 | 1 | - | 3 | 1 | 2 | 1 | 1 | 1 | - | - | 2 | - |
| 3M | 1 | - | 4 | 3 | 1 | - | 2 | 1 | 3 | 1 | 1 | 2 | - | - | 2 | - |
| 4M | 1 | - | 4 | 3 | 1 | 1 | 3 | 1 | 3 | 1 | 1 | 1 | - | - | 2 | - |

# I   Results of NoMod Analysis

As another performance metric for our approach, we measure the **NoMod** factor for the secrets/datasets we attack. Li et al. computed **NoMod** as follows: given a training dataset of LWE pairs ($\mathbf{Ra}$, $\mathbf{R}b$) represented

Table 22: **Average time (hours) to successful secret recovery with no model pre-training** ($n = 512, \log_2 q = 41$, binary secrets).

| # Samples | $h$ | | | | | | | | | | | | | | | |
|---|---|---|---|---|---|---|---|---|---|---|---|---|---|---|---|---|
| | 30 | 31 | 32 | 33 | 34 | 35 | 36 | 37 | 38 | 39 | 40 | 41 | 42 | 43 | 44 | 45 |
| 100K | - | - | - | - | - | - | - | - | - | - | - | - | - | - | - | - |
| 300K | - | - | 30.3 | - | - | - | - | - | - | - | - | - | - | - | - | - |
| 1M | - | - | 21.1 | 36.9 | 36.3 | - | 27.5 | 21.9 | 23.9 | 16.5 | 50.7 | 52.9 | - | - | 36.8 | - |
| 3M | 14.4 | - | 21.8 | 27.6 | 41.8 | - | 19.8 | 25.5 | 22.7 | 18.1 | 39.5 | 44.3 | - | - | 37.8 | - |
| 4M | 17.2 | - | 22.8 | 26.5 | 29.1 | 69.7 | 22.7 | 22.7 | 20.9 | 21.4 | 29.0 | 25.7 | - | - | 40.5 | - |

Table 23: **Successful secret recoveries out of 10 trials per $h$ with 80K steps model pre-training** ($n = 512, \log_2 q = 41$, binary secrets).

| # Samples | $h$ | | | | | | | | | | | | | | | |
|---|---|---|---|---|---|---|---|---|---|---|---|---|---|---|---|---|
| | 30 | 31 | 32 | 33 | 34 | 35 | 36 | 37 | 38 | 39 | 40 | 41 | 42 | 43 | 44 | 45 |
| 100K | - | - | 1 | 1 | - | - | - | - | - | - | - | - | - | - | - | - |
| 300K | 1 | - | 2 | 2 | - | - | 1 | - | 1 | 1 | - | - | - | - | 2 | - |
| 1M | 1 | - | 3 | 2 | 1 | - | 1 | - | 1 | 1 | 1 | - | - | - | 1 | - |
| 3M | 1 | - | 3 | 2 | 1 | - | 1 | - | 1 | 1 | 1 | - | - | - | 2 | - |
| 4M | 1 | - | 3 | 2 | - | - | 1 | - | 1 | 1 | 1 | - | - | - | 1 | - |

Table 24: **Average time (hours) to successful secret recovery with 80K steps model pre-training** ($n = 512, \log_2 q = 41$, binary secrets).

| # Samples | $h$ | | | | | | | | | | | | | | | |
|---|---|---|---|---|---|---|---|---|---|---|---|---|---|---|---|---|
| | 30 | 31 | 32 | 33 | 34 | 35 | 36 | 37 | 38 | 39 | 40 | 41 | 42 | 43 | 44 | 45 |
| 100K | - | - | 29.2 | 32.1 | - | - | - | - | - | - | - | - | - | - | - | - |
| 300K | 57.7 | - | 47.6 | 26.9 | - | - | 31.7 | - | 62.6 | 55.6 | - | - | - | - | 50.9 | - |
| 1M | 53.4 | - | 37.0 | 25.6 | 61.7 | - | 25.1 | - | 65.9 | 30.0 | 44.8 | - | - | - | 51.4 | - |
| 3M | 34.4 | - | 41.8 | 25.1 | 57.5 | - | 26.2 | - | 45.1 | 23.6 | 38.4 | - | - | - | 39.7 | - |
| 4M | 35.1 | - | 30.9 | 33.0 | - | - | 35.2 | - | 36.5 | 21.8 | 26.8 | - | - | - | 32.1 | - |

Table 25: **Successful secret recoveries out of 10 trials per $h$ with 240K steps model pre-training** ($n = 512, \log_2 q = 41$, binary secrets).

| # Samples | $h$ | | | | | | | | | | | | | | | |
|---|---|---|---|---|---|---|---|---|---|---|---|---|---|---|---|---|
| | 30 | 31 | 32 | 33 | 34 | 35 | 36 | 37 | 38 | 39 | 40 | 41 | 42 | 43 | 44 | 45 |
| 100K | - | - | - | 1 | - | - | - | - | - | - | - | - | - | - | - | - |
| 300K | 1 | - | 2 | 2 | - | - | 2 | - | 2 | 1 | - | - | - | - | - | - |
| 1M | 1 | - | 2 | 2 | 1 | - | 1 | - | 1 | 1 | 1 | - | - | - | 1 | - |
| 3M | 1 | - | 2 | 2 | - | - | 2 | - | 2 | 1 | 1 | - | - | - | 1 | - |
| 4M | 1 | - | 2 | 2 | 1 | - | 2 | - | 1 | 1 | 1 | - | - | - | 1 | - |

in the range $(-q/2, q/2)$ and known secret $\mathbf{s}$, compute $x = \mathbf{Ra} \cdot \mathbf{s} - \mathbf{R}b$. If $\|x\| < q/2$, we know that the computation of $\mathbf{Ra} \cdot \mathbf{s}$ did not cause $\mathbf{R}b$ to "wrap around" modulus $q$. The **NoMod** factor of a dataset is the percentage of $(\mathbf{Ra}, \mathbf{R}b)$ pairs for which $\|x\| < q/2$.

Although **NoMod** is not usable in a real world attack, since it requires a priori knowledge of $\mathbf{s}$, it is a useful metric for understanding attack success in a lab environment. Li et al. derived an empirical result stating that attacks should be successful when the **NoMod** factor of a dataset is $\geq 67$. The **NoMod** analysis

Table 26: **Average time (hours) to successful secret recovery with 240K steps model pre-training** ($n = 512, \log_2 q = 41$, binary secrets).

| # Samples | h | | | | | | | | | | | | | | | |
|---|---|---|---|---|---|---|---|---|---|---|---|---|---|---|---|---|
| | 30 | 31 | 32 | 33 | 34 | 35 | 36 | 37 | 38 | 39 | 40 | 41 | 42 | 43 | 44 | 45 |
| 100K | - | - | - | 21.0 | - | - | - | - | - | - | - | - | - | - | - | - |
| 300K | 49.3 | - | 30.8 | 23.3 | - | - | 50.5 | - | 64.2 | 37.9 | - | - | - | - | - | - |
| 1M | 49.4 | - | 29.5 | 23.4 | 50.8 | - | 20.1 | - | 24.8 | 24.2 | 51.9 | - | - | - | 61.9 | - |
| 3M | 48.9 | - | 38.2 | 19.7 | - | - | 38.5 | - | 50.9 | 20.2 | 57.7 | - | - | - | 47.4 | - |
| 4M | 42.4 | - | 34.9 | 19.5 | 55.8 | - | 29.0 | - | 24.1 | 18.3 | 54.0 | - | - | - | 52.4 | - |

Table 27: **Successful secret recoveries out of 10 trials per $h$ with 440K steps model pre-training** ($n = 512, \log_2 q = 41$, binary secrets).

| # Samples | h | | | | | | | | | | | | | | | |
|---|---|---|---|---|---|---|---|---|---|---|---|---|---|---|---|---|
| | 30 | 31 | 32 | 33 | 34 | 35 | 36 | 37 | 38 | 39 | 40 | 41 | 42 | 43 | 44 | 45 |
| 100K | - | - | 1 | 1 | - | - | - | - | - | - | - | - | - | - | - | - |
| 300K | - | - | 1 | 2 | - | - | 1 | 1 | 1 | 1 | - | - | - | - | 1 | - |
| 1M | 1 | - | - | 2 | - | - | 1 | - | 2 | 1 | 1 | - | - | - | 1 | - |
| 3M | 1 | - | 1 | 2 | - | - | 1 | - | 1 | 1 | 1 | - | - | - | 2 | - |
| 4M | 1 | - | 1 | 2 | - | - | 2 | - | 1 | 1 | 1 | - | - | - | 2 | - |

Table 28: **Average time (hours) to successful secret recovery with 440K steps model pre-training** ($n = 512, \log_2 q = 41$, binary secrets).

| # Samples | h | | | | | | | | | | | | | | | |
|---|---|---|---|---|---|---|---|---|---|---|---|---|---|---|---|---|
| | 30 | 31 | 32 | 33 | 34 | 35 | 36 | 37 | 38 | 39 | 40 | 41 | 42 | 43 | 44 | 45 |
| 100K | - | - | 33.3 | 20.0 | - | - | - | - | - | - | - | - | - | - | - | - |
| 300K | - | - | 39.8 | 12.9 | - | - | 29.0 | 60.5 | 36.2 | 49.1 | - | - | - | - | 59.1 | - |
| 1M | 56.9 | - | - | 16.6 | - | - | 18.6 | - | 43.6 | 24.9 | 36.4 | - | - | - | 53.4 | - |
| 3M | 50.9 | - | 32.3 | 12.2 | - | - | 25.2 | - | 22.3 | 18.9 | 42.4 | - | - | - | 60.8 | - |
| 4M | 58.3 | - | 30.0 | 13.5 | - | - | 34.6 | - | 19.7 | 22.3 | 45.7 | - | - | - | 59.7 | - |

indicates that models trained in those experiments were only learning secrets from datasets in which the majority of $\mathbf{R}b$ values do not "wrap around" $q$. If models could be trained to learn modular arithmetic better, this might ease the NoMod condition for success.

One of the main goals of introducing the angular embedding is to introduce some inductive bias into the model training process. Specifically, teaching models that $0$ and $q - 1$ are close in the embedding space may enable them to better learn the modular arithmetic task at the heart of LWE. Here, we examine the **NoMod factor** of various datasets to see if the angular embedding does provide such inductive bias. If it did, we would expect that models with angular embeddings would recover secrets from datsets with **NoMod** $< 67$. Table 29 lists **NoMod** percentages and successful secret recoveries for the angular and tokenization schemes described in §4.2.

Table 29: **NoMod percentages for Verde data** $n = 512, \log_2 q = 41$**, binary secrets (varying $h$ and secrets indexed 0-9), comparing performance of angular vs. normal embedding.** Key: **recovered by angular only**, *recovered by both*, not recovered.

| $h$ | 0 | 1 | 2 | 3 | 4 | 5 | 6 | 7 | 8 | 9 |
|---|---|---|---|---|---|---|---|---|---|---|
| 57 | 45 | 49 | 52 | 41 | 51 | 46 | 51 | **57** | **60** | 49 |
| 58 | 48 | 48 | 48 | 48 | 38 | 43 | 52 | 52 | 45 | 48 |
| 59 | 43 | 46 | **56** | 50 | *66* | *63* | 35 | 46 | 41 | 40 |
| 60 | 48 | 48 | 52 | **59** | 50 | **58** | 48 | 49 | 51 | 54 |
| 61 | **60** | 49 | 43 | 41 | **56** | 42 | 42 | 41 | 41 | 50 |
| 62 | 45 | 42 | 45 | 54 | 55 | 43 | **61** | 56 | 54 | 42 |
| 63 | 56 | **60** | 55 | 54 | *63* | 47 | 54 | 51 | 45 | 43 |
| 64 | 44 | 46 | 41 | 41 | 41 | 47 | 45 | 43 | 41 | 55 |
| 65 | 45 | 51 | 48 | **60** | 45 | 48 | 41 | 48 | 45 | 50 |
| 66 | 45 | 51 | 39 | **64** | 45 | 47 | 43 | **60** | 48 | 55 |
| 67 | 43 | 47 | 48 | 49 | 40 | 47 | 48 | 51 | 50 | 46 |

Table 30: **NoMod percentages for** $n = 768$, $\log_2 q = 35$ **secrets (varying $h$ and secrets indexed 0-9).** Key: **secret recovered**, secret not recovered.

| $h$ | 0 | 1 | 2 | 3 | 4 | 5 | 6 | 7 | 8 | 9 |
|---|---|---|---|---|---|---|---|---|---|---|
| 5 | 61 | 61 | 52 | **61** | **93** | **68** | **66** | 61 | 56 | 62 |
| 7 | 52 | 61 | 56 | **77** | 52 | **68** | 56 | **67** | 56 | 61 |
| 9 | 52 | 55 | 46 | 49 | 52 | 48 | 56 | 60 | 60 | **70** |
| 11 | 55 | 42 | 44 | 55 | 46 | 46 | 54 | 55 | 60 | 51 |
| 13 | 46 | 45 | 60 | 48 | 43 | 43 | 55 | 48 | 60 | 43 |
| 15 | 41 | 38 | 48 | 43 | 40 | 43 | 45 | 43 | 43 | 59 |

Table 31: **NoMod percentages for** $n = 1024$, $\log_2 q = 50$ **secrets (varying $h$ and secrets indexed 0-10).** Key: **secret recovered**, secret not recovered.

| $h$ | 0 | 1 | 2 | 3 | 4 | 5 | 6 | 7 | 8 | 9 |
|---|---|---|---|---|---|---|---|---|---|---|
| 5 | **62** | **66** | **70** | **69** | **81** | **81** | **81** | 53 | **69** | **94** |
| 7 | 62 | 47 | **80** | **80** | **69** | 56 | 57 | 52 | 57 | **69** |
| 9 | 56 | **69** | 52 | 49 | 52 | 53 | 56 | 57 | 46 | 52 |
| 11 | 44 | 52 | 52 | 62 | **61** | 62 | 56 | 56 | 52 | 49 |
| 13 | 51 | 46 | 56 | 40 | 46 | 52 | 44 | 49 | **68** | 53 |
| 15 | 61 | 46 | 40 | 45 | 46 | 41 | 38 | 47 | 48 | 44 |

