# OpenReview forum: "Salsa Fresca: Angular Embeddings and Pre-Training for ML Attacks on Learning With Errors"
_TMLR — Accepted by TMLR_

### Review · Reviewer_FZQP · 2024-12-21

**Summary Of Contributions:**

The paper builds upon existing machine learning-based attacks on the Learning with Errors (LWE) problem by leveraging off-the-shelf cryptographic tools (e.g., Flatter), and a novel algorithmic technique (e.g., angular embedding). These engineering tweaks substantially speed up the pre-processing phase, and enable the attack to scale effectively to real-world LWE parameters, such as $n$=1024, commonly employed in homomorphic encryption schemes.

**Audience:**

Yes

**Broader Impact Concerns:**

Addressed in the paper

**Claims And Evidence:**

Yes

**Requested Changes:**

See the weakness.

**Strengths And Weaknesses:**

## Strengths

$\bullet$ The authors address a highly challenging computational problem, and their pre-processing improvements play a crucial role in making such attacks computationally feasible. If successful, these attacks could significantly impact the security of post-quantum cryptography (PQC), underscoring the importance of this work in evaluating the robustness and security guarantees of PQC schemes.

$\bullet$ The paper provides a solid empirical evaluation, with detailed comparisons to previous ML-based approaches for LWE attacks. The authors have done a commendable job of thoroughly discussing the attack parameters and hyperparameters, which adds to the transparency and reproducibility of the work.

$\bullet$ The paper is well-organized and written clearly and concisely. Even readers who are not experts in cryptography will find it easy to follow.


## Weaknesses


$\bullet$ The paper lacks deeper insights and novel algorithmic contributions. While these engineering tweaks offer significant speedup, the authors could consider exploring and highlighting any insights that may have emerged during their work.


$\bullet$ The paper does not quantitatively or qualitatively demonstrate the advantages of using sub-quadratic or linear softmax-based attention mechanisms (see [1,2,3]) in terms of speeding up the attacks.

It would strengthen the work if the authors could include specific experiments or analyses that clearly show the impact of these mechanisms on computational efficiency and attack performance.

$\bullet$ The authors should include a detailed breakdown of the speedup improvements, demonstrating how specific components contribute to the overall improvement (e.g., achieving a 25$\times$ speedup for $n$=512). This analysis would provide a clearer understanding of the most impactful factor in achieving the reported performance gains, and strengthen the empirical evaluation.

[1] Zhang et al., The Hedgehog & the Porcupine: Expressive Linear Attentions with Softmax Mimicry, ICLR 2024

[2] Zhang et al., Gated Slot Attention for Efficient Linear-Time Sequence Modeling, NeurIPS 2024.

[3] Han at al., Bridging the Divide: Reconsidering Softmax and Linear Attention, NeurIPS 2024.

---

> ### Author Response · Authors · 2025-01-29
> **Rebuttal**
>
> Thank you for your thorough and constructive review. We appreciate your recognition of the paper's strengths and would like to address your specific concerns:
>
> **Deeper Insights Beyond Engineering**
>
> While our improvements may appear primarily engineering-focused, we believe there are significant insights worth highlighting more prominently in our revision:
>
> 1. The surprising composability of our optimizations is noteworthy. In ML, it's common that combining multiple 2x improvements does not yield the expected multiplicative speedup---yet our improvements across preprocessing (25x), architecture (25% more samples/sec), embeddings (2.4x faster), and pre-training (10x fewer samples) do compose effectively. This suggests something fundamental about the structure of the LWE problem that allows independent optimizations to stack multiplicatively.
>
> 2. Our angular embedding's success demonstrates how incorporating problem-specific structure (in this case, the modular nature of LWE) into neural architectures can yield significant improvements - a principle that may generalize to other domains with similar mathematical structures.
>
> **Attention Mechanisms**
>
> While we appreciate the pointer to recent work on efficient attention mechanisms, we believe exploring alternative attention mechanisms is outside the scope of our work. Our paper focuses on the fundamental challenge of applying ML to the LWE problem - specifically, how to effectively represent and learn from modular arithmetic data. The computational efficiency of the attention mechanism, while important, is secondary to these core representational challenges. Our angular embedding approach already reduces sequence length from $2n$ to $n$ tokens, which provides sufficient computational benefits for our purposes. More importantly, this reduction stems from better problem representation rather than architectural optimization.
>
> **Speedup Breakdown**
>
> We actually provide a detailed breakdown of our speedup contributions in the conclusion:
> - 250x fewer CPU hours spent preprocessing
> - 3x more samples/sec for n=512
> - 2.4x faster training from angular embeddings
> - 10x sample reduction from pre-training
>
> These improvements compound to enable the first successful ML attack on larger parameter sets ($n=768$, $n=1024$).
>
> We will reorganize this information to make it more prominent in the paper, helping readers better understand the relative impact of each component.
>
> Thank you for the thoughtful suggestions that will help us improve the presentation of our work.

---

> > ### Comment · Reviewer_FZQP · 2025-02-03
> > **Response to Authors' Rebuttal**
> >
> > Thank you for the rebuttal!
> >
> > I appreciate the authors' detailed breakdown of efficiency gains as it clearly explains the key factors behind the speedup.  I would recommend adding a visual representation, such as a pie chart in the introduction, along with a brief discussion to highlight the impact of ML contributions (e.g., angular embeddings) on the efficiency of ML-based LWE attacks.
> >
> > Also, I  would suggest adding a discussion on promising ML techniques, such as linear softmax, that have the potential to further improve the efficiency of LWE attacks, along with their broader implications. This could provide deeper insights into future directions in this space

---

### Review · Reviewer_hCFg · 2025-01-09

**Summary Of Contributions:**

This paper focuses on improving machine learning (ML) attacks on the Learning with Errors (LWE) problem underlying post-quantum cryptography with three major techniques: better preprocessing, angular embeddings and model pre-training. The experiments show that the proposed method can improve the efficiency of the attacks on LWE problems.

**Audience:**

Yes

**Claims And Evidence:**

Yes

**Requested Changes:**

1. Can authors further explain the motivation behind the proposed angular embedding? What are the advantages of the proposed angular embedding compared to the previous models? How does the angular embedding can overcome the shortcomings, namely quadratic computational cost and lose of inductive bias of number continuity.

2. What is the computational expense of model pre-training? When model pre-training is included, is it still more efficient than the approach without model pre-training?

**Strengths And Weaknesses:**

Strengths:
1. The paper is well organized and easy to follow.
2. The method is well illustrated.
3. The experimental results show promising results.

Weaknesses:
1. Some of the contributions claimed by the authors seem more like technical improvements rather than strict novel methods. For example, the methods adopted in the pre-processing phase are off-the-shelf.
2. The paper proposed angular embedding in the transformer. but the motivation is not clearly justified. Besides, the advantages of the proposed angular embedding over the previous models are not fully discussed.
3. From Figure 5, it seems that the pre-training could not effectively reduce the mean hours on large training dataset (e.g., 3M and 4M).

---

> ### Author Response · Authors · 2025-01-29
> **Rebuttal**
>
> Thank you for the time and effort spent reading and engaging with our work. Your feedback is helpful and we appreciate it. We’d like to discuss both the weaknesses you found and your requested changes.
>
> Weaknesses:
> 1. While we agree that we apply existing work, there are interesting findings that we perhaps do not explicitly discuss. Please see the general discussion on optimizations composing well.
> 2. We discuss motivation and additional analysis of angular embeddings in the general response above.
> 3. We agree and thank you for the careful reading. This is not unexpected: with sufficient downstream task samples, pre-training is not useful. For example, BERT is commonly used in NLP for an example sentiment analysis task. With only 100 samples, BERT will massively outperform a randomly initialized network. However, given millions of labeled examples for a sentiment analysis task, a randomly initialized network is likely to perform on par with BERT. We will add a discussion of this and present it as an expected limitation in an explicit, newly-added limitations section in the final version.
>
> Your requested changes:
>
> 1. We will add additional discussion of the angular embeddings to the final version; we have also added some discussion in the general response. But to answer your specific questions:
>
>
> What are the advantages of the proposed angular embedding compared to previous models?
>
> Previous embeddings have two limitations. Quoting from our paper:
>
> > First, input sequences are $2n$ tokens long, which slows training as n grows, because transformers’ attention mechanism scales quadratically in sequence length.
>
> > Second, all vocabulary tokens are learned independently and the inductive bias of number continuity is lost.
>
> $2n$ sequence length and a lack of inductive bias are the weaknesses in previous model embedding strategies (vocabulary based embeddings, like language models).
>
> We argue that we can solve both of these by introducing a better embedding technique.
>
> > To address these shortcomings, we introduce an angular embedding which strives to better represent the problem’s modular structure in embedding space, while encoding integers with only one token.
>
> While the angular embedding does not solve the quadratic complexity, it does halve the sequence length with respect to $n$: $2n$ tokens with vocabulary-based embeddings -> $n$ tokens with angular embeddings.
>
> It also solves the number continuity problem. Since all numbers are represented as 2-D coordinates on the unit circle, with 0 and $q$ at (0, 1), $q/2$ at (0, -1), $q/4$ at (1, 0), etc, then applying the same linear projection to all numbers ($W_e$)  leads to a ellipsis in the model embedding space.
>
> 2. Recovering a single secret is more expensive with pre-training when including pre-training. However, the same pre-trained checkpoint can recover multiple, different, unknown secrets. Thus, the pre-training cost can be spread out over many secret recoveries. Furthermore, as we discuss above, pre-training is motivated by reducing the number of required samples, which leads to the majority of improvements in total secret recovery time.

---

> ### Comment · Reviewer_hCFg · 2025-02-10
> **Response**
>
> Thanks for the rebuttal. After reviewing the replies and comments from other reviewers, I believe this paper makes a meaningful contribution to the ML community by integrating ML techniques into cryptographic problems. Specifically, it demonstrates a significant efficiency improvement in solving the LWE problem using both off-the-shelf techniques (e.g., pre-training) and the proposed method (e.g., angular embedding), which could inspire future research in this area. Given this, I would like to update the Claims and Evidence section from "No" to "Yes."
>
> Meanwhile, I still have a question regarding the angular embedding. How does the proposed embedding technique halve the length of the input sequence? Could the authors provide more details on this?

---

> > ### Comment · Action_Editor_fpwC · 2025-02-18
> >
> > Dear Authors,
> >
> > Would you like to respond to this query?
> >
> > Regards,
> > AE

---

> > ### Author Response · Authors · 2025-02-21
> > **Input Sequence Length**
> >
> > TL;DR: The token-based embedding (from prior work) uses 2 tokens per element of $\mathbf{a}$: if $n = 512$, then the transformer has $1024$ input tokens. Our method uses 1 token per element of $\mathbf{a}$: if $n=512$, then the transformer sees $512$ input tokens.
> >
> >
> > **Details:** As we mention in Section 4.2, there are $q$ possible inputs for each element of $\mathbf{a}$ because $\mathbf{a} \in \mathbb{Z}_q^n$. However, the smallest $q$ we attack is $\ge 2^{35}$. That's an infeasibly large vocabulary size for a transformer. Prior work solves this problem by splitting $q$ into $\frac{q}{B}$ and $q \mod B$ such that the vocabulary size is smaller than 10K (they also use binning on $q \mod B$ but that's irrelevant for our purposes). However, each element of $\mathbf{a}$ now needs to be represented with two tokens: $\frac{q}{B}$ and $q \mod B$. Thus, for $n=512$, the transformer sees $1024$ tokens.
> >
> > We instead represent every element of $\mathbf{a}$ as a position on the unit circle. Although this relies on floating point precision, we find that in practice, the magnified error as a result of lattice reduction is much larger than any errors introduced by floating point precision.

---

> > > ### Comment · Reviewer_hCFg · 2025-02-24
> > > **Reply**
> > >
> > > Thanks for your clarification.

---

### Review · Reviewer_YQZC · 2025-01-14

**Summary Of Contributions:**

The paper proposes a set of training methods to improve computational performance of the learning with errors problem in post-quantum cryptography. The authors present a significant improvement in computational time through the combining of established pre-processing techniques and pre-training methods. Angular embeddings are also introduced to improve recovery results, with the authors stating such embeddings more naturally align to the problem setting and domain. The authors provide a detailed empirical analysis and a number of ablations to demonstrate the improvements of their method and achieve recovery that has yet to be seen in existing literature.

**Audience:**

Yes

**Broader Impact Concerns:**

The broader impact statement could be expanded further. There is a significant number of ethical risks that implicate this work, and the current statement could elaborate further.

**Claims And Evidence:**

Yes

**Requested Changes:**

1. Focus the writing more on the ML contributions and motivation. While the problem domain is well defined, more efforts should be made to adhere to the focus of the venue. For example, the introduction only briefly discusses the ML challenges and motivation.
2. More focus on the motivation, and analysis of the angular embeddings, this is the primary contribution of the work, and is reduced to a single paragraph, that does not describe well its context.

**Minor Changes**:
1. Table 1, clarify what has the attack time been reduced from? You state it is now 50 from which value.
2. Re-work section 3 to make it clear flutter is a previous development, utilised by the authors. In addition, provide quantitative values for the experimental setup of section 3. For example, replace “many” with specific values to aid reproduction.

**Strengths And Weaknesses:**

**Strengths**
1. Introduces the LWE and PQC problems well to non-expert readers.
2. The angular embedding is an interesting development that has empirical performance improvements over a standard vocabulary embedding.
3. While the approach in and of itself is limited in its novelty, the combination of methods demonstrates tangible improvements and presents a significant contribution to the field of LWE.
4. The number of experimentation and ablations demonstrate well the component based performance and trade-offs.
5. The work is generally well written and presented.

**Weaknesses**
1. My biggest concern is the scope and positioning of the paper within Machine Learning research. I would arguably state that this method would be better placed in security focussed venue, given the limited contribution to the ML field of which TMLR is positioned.
2. Much of the advancements do not lie in the ML research space. Rather, a number of existing methods have been combined to produce a more innovative systems approach to the application domain. Hence, for this venue the novelty is limited.
3. The angular embedding section defines the development well, however, the choice is not well motivated. For example, how does the angular embeddings help represent the modular structure of the problem. A few more questions regarding the angular embedding are given below:
    a. How does the learned linear projection W_e ensure an ellipse is learnt?
    b. Did you try a vocabulary embedding with the additional learnable linear layers W_e and W_a? If so, does this improve the performance of vocabulary embeddings?
    c. More analyses of the embeddings would be desirable to ensure the expected behaviour is being observed.
4. Limitations of the work are not discussed.

**Minor Comments**
1. It is not clear how or why the problem setting was defined, why the values in table 3, why not alternative values or other settings. Does this follow prior works?
2. It is not clear in the text if Flatter has been developed in this work or has been utilised by the authors. I assume the former as this has not been described in detail or anonymised.
3. The addition of “Salsa Fresca” in the title does not help describe the work or its contributions, and perhaps an abbreviation of the methods used would be better placed.

---

> ### Author Response · Authors · 2025-01-29
> **Rebuttal**
>
> Thank you for your time and effort spent reading and engaging with our work. We appreciate the detailed review and specific technical questions. We addressed concerns about technical novelty, relevance to a machine learning journal like TMLR and discussion on angular embeddings above in our general rebuttal.
>
> We’d like to address your specific questions:
> 1. Scope and positioning: we discuss in the general rebuttal, but we firmly believe the work is a good fit for TMLR.
> 2. Advancements in ML: we discuss in the general rebuttal, but we firmly believe our contributions are highly relevant for ML researchers.
> 3. Motivating the angular embedding: we discuss in the general rebuttal, but we will add further discussion in Section 4 and additional intuition in the Appendix.
> 4. With respect to limitations: we will add an explicit “Limitations” section to the appendix. In short, our method is limited by two factors, described below. We will add further, in-depth discussion to the final version of our work.
>     1. How effectively lattice reduction methods reduce samples. Our work integrates and leverages the latest developments in lattice reduction to improve the number of samples with scalar products a * s that lie in (-q/2, q/2); this property is referred to as NoMod in our work and prior work and is discussed in the last paragraph of Section 2.4.
>     2. The ability of the transformer to learn from unreduced samples. We do not focus on this component in our work and leave it to future work; we will include this discussion in the final version.
>
> Your minor comments:
>
> 1. Regarding Table 3 parameter settings: These values were chosen to align with real-world post-quantum cryptography implementations, specifically trying to achieve the parameter sets from real-world specifications for parameters. Please see Section 2.4 for additional discussion of the parameters used in our work.
> 2. About Flatter: We apologize for the ambiguity. Flatter is indeed a previously developed tool [1] and is cited in our work. We will, however, revise Section 3 to make this clearer.
> 3. Regarding the title: We appreciate the feedback about "Salsa Fresca"; we chose it to follow up on previous work in the SALSA line of work [2, 3, 4], as noted in the related work (Appendix C).
>
> Your requested changes:
>
> 1. While we firmly believe that improving applications of ML to problems of interest are worth publishing, we are happy to add additional discussion and motivation from an ML perspective. There are interesting challenges in trying to find a reliable gradient signal in a modular problem, as evidenced by [5, 6, 7].
> 2. We disagree that angular embeddings are the primary contribution of this work. We apply a spread of optimizations that reduce required CPU hours by 250x and increase transformer samples/sec by 3x. However, we are happy to add more discussion and motivation for angular embeddings.
>
> And your minor changes:
>
> 1. Table 1: as we state in the main text, prior work has not successfully recovered secrets for the settings we consider. However, Section 6 extensively compares our work with other attacks to give a broader context to readers.
> 2. Thank you for the feedback. We will make it clear that Flatter is a previously developed tool that we leverage. We will also add specific counts instead of the word “many” throughout that section.
>
> [1] Keegan Ryan and Nadia Heninger. Fast practical lattice reduction through iterated compression. Cryptology ePrint Archive, 2023. URL https://eprint.iacr.org/2023/237.pdf.
>
> [2] Emily Wenger, Mingjie Chen, Francois Charton, and Kristin Lauter. Salsa: Attacking lattice cryptography with transformers. In Proc. of NeurIPS, 2022.
>
> [3] Cathy Yuanchen Li, Jana Sotáková, Emily Wenger, Mohamed Malhou, Evrard Garcelon, François Charton, and Kristin Lauter. Salsa Picante: A Machine Learning Attack on LWE with Binary Secrets. In Proc. of ACM CCS, 2023a.
>
> [4] Cathy Yuanchen Li, Emily Wenger, Zeyuan Allen-Zhu, Francois Charton, and Kristin E Lauter. SALSA VERDE: a machine learning attack on LWE with sparse small secrets. In Proc. of NeurIPS, 2023b
>
> [5] Li, Chenyang, et al. "Fourier circuits in neural networks and transformers: A case study of modular arithmetic with multiple inputs." arXiv e-prints (2024): arXiv-2402.
>
> [6] Gromov, Andrey. "Grokking modular arithmetic." arXiv preprint arXiv:2301.02679 (2023).
>
> [7] Mallinar, Neil, et al. "Emergence in non-neural models: grokking modular arithmetic via average gradient outer product." arXiv preprint arXiv:2407.20199 (2024).

---

> > ### Comment · Reviewer_YQZC · 2025-02-03
> > **Response to Rebuttal**
> >
> > I thank the authors for their rebuttal, and their clarification of some points.
> >
> > Response 1-3 of the specific question should be more concretely justified, as there has been no additional information to support the contrary claim by the authors. I do however appreciate the clarification of the limitations and the minor comments.
> >
> > I however, still stand by my previous points that this paper is not well suited to the TMLR venue given the limited contributions to the ML field. Firstly, the rebuttal does not present any further information or justification that concretely describes the reasoning for the angular embeddings, how does it specifically suit the problem setting? it is not clear. I would also stand by my point that this is the primary contribution of this work, thus should be presented as such. As mentioned by other reviewers, pertaining is a standard approach and not a contribution in the ML setting, rather one for the application domain (not a negative comment just related to the framing of the method), and the speedup has been achieved through the implementation of existing tools, not a distinct contribution by the authors.
> >
> > The above points do not discredit the work, the overall system is interesting and clearly a contribution to the security domain, however, it has not been clarified how its application and contributions add to the machine learning community.

---

### Author Response · Authors · 2025-01-29
**General Discussion**

**ML Contributions and Venue Fit (YQZC, hCFg)**

We appreciate the concerns about the ML contributions and venue fit. While our work indeed has security implications, our core contributions are fundamentally ML-focused:

1. The angular embedding is a novel neural architecture component that explicitly addresses the limitations of standard embeddings for numerical data in transformer models.
2. Our pre-training approach demonstrates how to effectively leverage domain knowledge in self-supervised learning for cryptographic problems.
3. The improved preprocessing showcases how ML pipelines can be optimized for specific problem domains.

**Angular Embedding Motivation and Analysis (YQZC, hCFg)**

We thank the reviewers for pushing us to better motivate and analyze the angular embedding. The embedding was specifically designed to exploit the cyclic/modular structure of LWE problems---standard vocabulary embeddings lack this inductive bias between values.
Angular embeddings preserve the circular structure present in modular problems. By first representing integers in $\mathbb{Z}_q$ as 2D points on the unit circle, we preserve the modular structure of the problem such that 0 is equally close to both 1 and q-1. We train only a single linear projection $W_e$ from $\mathbb{R}^2$ to $\mathbb{R}^D$, which can only lead to an ellipsis in the higher dimensional space.

We will further clarify how angular embeddings preserve elliptical structure in the embedding space in the revised version.

**Pre-training Efficiency (hCFg, FZQP)**

Regarding the computational expense of pre-training and its benefits at scale:

1. While pre-training does add upfront cost, it is amortized across multiple attack instances. The same pre-trained checkpoint can be used to recover multiple, different secrets.
2. The apparent diminishing returns at larger scales (3-4M) warrant further discussion - we will add analysis of the cost-benefit trade-off.
3. We argue that pre-training improves the *sample efficiency* of secret recovery—rather than needing 4M samples, we can recover secrets with as few as 300K samples. Because the majority of our secret recovery cost is in finding reduced samples (25K CPU hours vs 36 GPU hours), methods that reduce the number of needed samples are of huge importance. Pre-training thus leads to massive gains for secret recovery.

We should emphasize the surprising nature of our optimizations composing well. In many ML settings, a 2x improvement from a novel optimizer and a 2x improvement from better data does not lead to a 4x improvement. In an interesting twist, our improvements across the entire pipeline *do* compose.

Quoted from our conclusion:

> Our contributions are spread across multiple fronts: faster preprocessing (25x fewer CPU hours), simpler architecture (25% more samples/sec), better token embeddings (2.4x faster training) and the first use of pre-training for LWE (10x fewer samples). These lead to 250x fewer CPU hours spent preprocessing and 3x more samples/sec for $n = 512$, $\log_2 q = 41$ LWE problems, and lead to the first ML attack on LWE for $n = 768$ and $n = 1024$.

We will add additional discussion of the surprising nature of combining optimizations.

---

Generally, we would like to emphasize how our work aligns with TMLR's acceptance criteria. Our empirical evidence clearly demonstrates quantified improvements: 250x fewer CPU hours for preprocessing, 3x more samples/sec for $n=512$, 2.4x faster training from angular embeddings, and 10x fewer samples needed with pre-training. While focused on a specific application, our findings about combining architectural improvements, domain-specific embeddings, and pre-training approaches offer valuable insights for ML researchers working with structured mathematical problems. These results demonstrate both the validity of our claims and their relevance to TMLR's audience.

---

### Decision · Action_Editor_fpwC · 2025-02-24

**Recommendation:** Accept with minor revision

**Comment:**

In this paper, the authors introduce three key improvements to machine learning attacks on the learning with errors problem: **a)** faster preprocessing, **b)** angular embeddings, and **c)** model pre-training, which make ML attacks on LWE more efficient and scalable.

By integrating the Flatter algorithm and interleaving BKZ and polishing techniques, preprocessing is accelerated by 25×, while an encoder-only transformer architecture with angular embeddings reduces computational complexity, improving training efficiency. Additionally, the paper explores pre-training for ML-based LWE attacks, reducing the number of training samples required by 10×, making attacks significantly more sample-efficient.

The reviewers have generally been positive but have highlighted areas that could improve the paper and should be considered carefully when preparing the camera-ready version, given that the authors did not submit a revised version during the reviewer-author discussion. Therefore, the statements that authors have made during their responses need to be reflected in the camera-ready version submission. For example:

**a)** Provide a clearer motivation for angular embeddings and how they preserve cyclic structure of LWE problems

**b)** Add additional discussion on “the surprising nature of combining optimizations”

**c)** Incorporate the references and statements made in response to reviewer YQZC, e.g. around the Flatter Algorithm referencing

**d)** Expand section 2.4 to justify why specific LWE parameters were chosen

**e)** Please reorganise the information mentioned in response to FZQP reviewer related to “Speedup Breakdown”

**Audience:**

Yes - the paper will appeal to at least part of TMLR's audience, especially in the intersection of security and machine learning.

**Claims And Evidence:**

Yes - the reviewers agree that the algorithmic, methodological, and empirical evidence presented support the main claims of the paper.

For instance, the authors demonstrate that their improvements led to the first successful ML-based attack on sparse binary secrets for n = 1024, the smallest dimension used in practice for homomorphic encryption applications of LWE. In addition, the model achieves 3× more samples per second and 2.4× faster training through architecture refinements.

Their proposed model learns reusable patterns, allowing multiple attacks with fewer new samples, cutting the total compute cost of secret recovery. Their empirical results also show that angular embeddings reduce sequence length, halving token requirements while maintaining accuracy.

---

> ### Author Response · Authors · 2025-03-24
> **Camera-ready submitted**
>
> We have submitted a camera-ready version with the requested changes.